# Beyond accuracy: generalization properties of bio-plausible temporal credit assignment rules

Yuhan Helena Liu[1,2,3,*], Arna Ghosh[4,5], Blake A. Richards[4,5,6,7], Eric Shea-Brown[1,2,3], and Guillaume Lajoie[5,7,8,*]

[1]Department of Applied Mathematics, University of Washington, Seattle, WA, USA
[2]Allen Institute for Brain Science, 615 Westlake Ave N, Seattle WA, USA
[3]Computational Neuroscience Center, University of Washington, Seattle, WA, USA
[4]School of Computer Science, McGill University, Montreal, QC, Canada
[5]Mila - Quebec AI Institute, Montreal, QC, Canada
[6]Department of Neurology and Neurosurgery, Montreal Neurological Institute, McGill University, Montreal, QC, Canada
[7]Canada CIFAR AI Chair, CIFAR, Toronto, ON, Canada
[8]Dept. de Mathématiques et Statistiques, Université de Montréal, Montreal, QC, Canada
[*]Correspondence: hyliu24@uw.edu, g.lajoie@umontreal.ca

## Abstract

To unveil how the brain learns, ongoing work seeks biologically-plausible approximations of gradient descent algorithms for training recurrent neural networks (RNNs). Yet, beyond task accuracy, it is unclear if such learning rules converge to solutions that exhibit different levels of generalization than their non-biologically-plausible counterparts. Leveraging results from deep learning theory based on loss landscape curvature, we ask: how do biologically-plausible gradient approximations affect generalization? We first demonstrate that state-of-the-art biologically-plausible learning rules for training RNNs exhibit worse and more variable generalization performance compared to their machine learning counterparts that follow the true gradient more closely. Next, we verify that such generalization performance is correlated significantly with loss landscape curvature, and we show that biologically-plausible learning rules tend to approach high-curvature regions in synaptic weight space. Using tools from dynamical systems, we derive theoretical arguments and present a theorem explaining this phenomenon. This predicts our numerical results, and explains why biologically-plausible rules lead to worse and more variable generalization properties. Finally, we suggest potential remedies that could be used by the brain to mitigate this effect. To our knowledge, our analysis is the first to identify the reason for this generalization gap between artificial and biologically-plausible learning rules, which can help guide future investigations into how the brain learns solutions that generalize.

## 1   Introduction

A longstanding question in neuroscience is how animals excel at learning complex behavior involving temporal dependencies across multiple timescales and thereafter generalize this learned behavior to unseen data. This requires solving the temporal credit assignment problem: how to assign the contribution of past neural states to future outcomes. To address this, neuroscientists are increasingly turning to the mathematical framework provided by recurrent neural networks (RNNs) to model learning mechanisms in the brain [1, 2]. Temporal credit assignment in RNNs is typically achieved by backpropagation through time (BPTT), or other gradient descent-based optimization algorithms,

36th Conference on Neural Information Processing Systems (NeurIPS 2022).

none of which are biologically-plausible (or bio-plausible for short). Therefore, the use of RNNs as a framework to understand the computational principles of learning in the brain has motivated an influx of bio-plausible learning rules that approximate gradient descent [1–3].

The performance of such rules is typically quantified by accuracy. Although the accuracy achieved by these rules is often comparable to true gradient descent, little is known about the breadth of the emergent solutions, namely how well they generalize. Broadly speaking, generalization refers to a trained model's ability to adapt to previously unseen data, and is typically measured by the so-called **generalization gap**: the difference between training and testing error. This is especially important when learning complex tasks with nonlinear RNNs where the loss landscape is non-convex, and therefore, many solutions with comparable training accuracy can exist. These solutions, characterized as (local) minima in the loss landscape, can nonetheless exhibit drastically different levels of generalization (Figure 1). It is not clear if gradient-based methods like BPTT and the existing bio-plausible alternatives have a different tendency to converge to loss minima that provide better or worse generalization.

While the search for better predictors of the generalization performance remains an open issue in deep learning research [4], recent extensive studies identify flatness of the loss landscape at the solution point as a promising predictor for generalization [5–9]. Leveraging these empirical and theoretical findings, **we ask**: how do proposed biologically-motivated gradient approximations affect the flatness of the converged solution on loss landscape, and thereby, generalization?

Our overarching goal is to investigate generalization trends for existing bio-plausible temporal credit assignment rules, and in particular, examine how truncation-based bio-plausible gradient approximations can affect such trends. Specifically, **our contributions** are summarized as follows:

- In numerical experiments, we demonstrate across several well-known neuroscience and machine learning benchmark tasks that state-of-the-art (SoTA) bio-plausible learning rules for training RNNs exhibit worse and more variable generalization gaps, compared to true (stochastic) gradient descent (Figure 2A-C).

- Using the same experiments, we show that bio-plausible learning rules tend to approach high-curvature regions in synaptic weight space as measured by the loss' Hessian eigenspectrum (Figure 2D-F). Further, we verify that this correlates with worse generalization (Figure 2D-F and 3), which is consistent with the literature.

- We present a theorem to explain this phenomenon by examining the weight update equation as a discrete dynamical system, which sheds light on a potential connection between gradient alignment and the preference over converging to narrower minima (Theorem 1, Figure 4).

Given the core components in designing artificial neural networks: data, objective functions, learning rules and architectures [10], we investigate different learning rules while holding the data, objective function and architecture constant. SoTA RNN learning rules investigated include a three-factor rule with symmetric feedback (symmetric e-prop [11]), a three-factor rule using random feedback weights (RFLO [12]) and a multi-factor rule using local modulatory signaling (MDGL [13]). For an in-depth explanation of how these rules are implemented and why they are bio-plausible, please refer to Appendix A.2. We also encourage the reader to visit Appendix B for Theorem 1 proof and discussion on loss landscape geometry. In the last paragraph of the Discussion section, we discuss potential remedies implemented by the brain and provide preliminary results (Appendix Figure 5). To our knowledge, our analysis is the first to highlight and quantitatively provide a mechanistic explanation of the reason for this gap in solution quality between artificial and bio-plausible learning rules for RNNs, thereby motivating further investigations into how the brain learns solutions that generalize.

## 2 Related works

### 2.1 Bio-plausible gradient approximations

Investigating bio-plausible learning rules is of interest both to identify more efficient training strategies for artificial networks and to better understand learning in the brain [2, 3, 14–43]. In order for learning to reach a certain goal quantified by an objective, learning algorithms often minimize a loss function [10]. The error gradient, if available, tells us how each parameter should be adjusted in

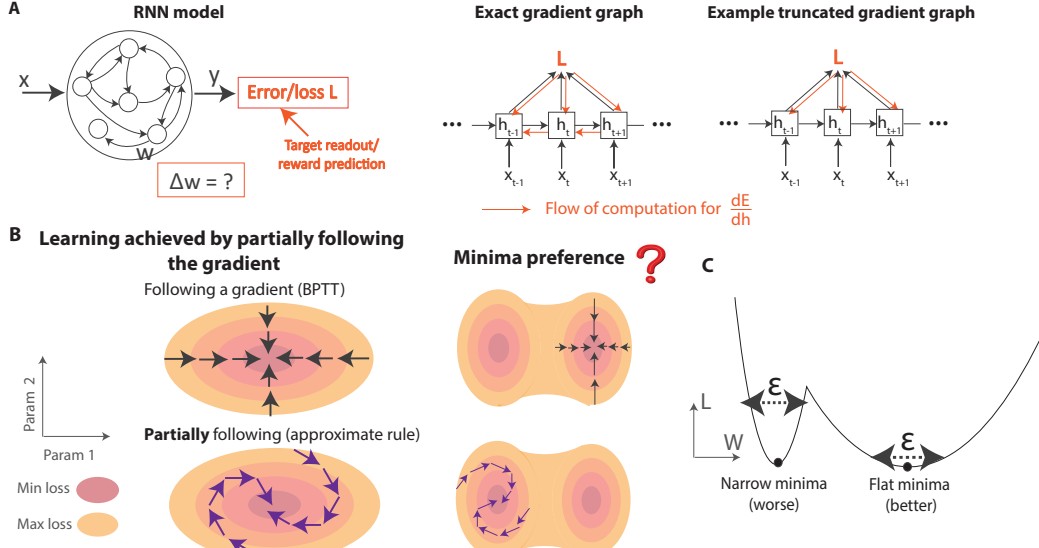

Figure 1: **Setup**. A) Illustration of an RNN trained to minimize error/loss function L (left). Existing bio-plausible proposals for RNNs estimate the gradient by neglecting dependencies that are biologically implausible to compute (right). B) Low training error/loss can be achieved by **partially** following a gradient (right), but the preference for converging to minima with certain generalization properties remains underexamined for these learning rules (left). C) Minima flatness matters: 1-D loss landscape illustration with two solutions that equally minimize loss $L$, but exhibit drastically different generalization properties: the narrower minima are more sensitive to perturbation.

order to have the steepest local descent. For training RNNs, which are widely used as a model for neural circuits [44–63], standard algorithms that follow this gradient — real time recurrent learning (RTRL) and BPTT — are not bio-plausible and have overwhelming memory storage demands [3, 64]. However, learning rules that only approximate the true gradient can sometimes be as effective as those that follow the gradient exactly [10, 65]. Because of that, bio-plausible learning rules that approximate the gradient using known properties of real neurons have been proposed and led to successful learning outcomes across many different tasks in feedforward networks [1, 66–75], with recent extensions to recurrently connected networks [11–13]. These existing bio-plausible rules for training RNNs [11–13] are truncation-based (which is the focus of this study), so that the untruncated gradient terms can be assigned with putative identities to known biological learning ingredients: eligibility traces, which maintain preceding activity on the molecular levels [76–81], combined with top-down instructive signaling [76, 77, 82–88] as well as local cell-to-cell modulatory signaling within the network [13, 89, 90]. For efficient online learning in RNNs, other approximations (not necessarily bio-plausible) to RTRL [91–96] have also demonstrated to produce a good performance. Given the impressive accuracy achieved by these approximate rules, several studies began to investigate their convergence properties [97], e.g. for random backpropagation weights in feedforward networks [98, 99]. However, the trend of generalization capabilities of these rules, especially in RNNs, is underexamined.

## 2.2 Loss landscape curvature and generalization performance

Given the central importance of understanding how neural networks perform in situations unseen during training [4, 100–106], the deep learning community has made tremendous efforts to develop tools for understanding generalization that we leverage here. That flat minima could lead to better generalization was observed more than two decades ago [107]. Intuitively, under the same amount of perturbation $\epsilon$ in parameter space (e.g. loss landscape changes due to the addition of new data) worse performance degradation will be seen around the narrower minima (see Figure 1C). We note that perturbations in parameter space can be linked to that in the input space [6]. Recently, many empirical studies have consistently supported the usefulness of this predictor [108–115]. In particular, the authors of [5] performed an extensive study and found that flatness-based measures have a higher correlation with generalization than other alternatives. Motivated by this, several studies have

characterized properties of the loss functions's Hessian — whose eigenspectrum carries information about curvature [116–120]. Connections between flatness and generalization performance have shed light on the reason for greater generalization gaps in large batch training [111, 121–124], and also have inspired optimization methods to favor flatter minima [108, 109, 125–132]. Despite the criticism of scale-dependence of flatness [133], where parameter rescaling can drastically change flatness but not always generalization quality, flatness — with parameter scales taken into account [134, 135] — are connected to PAC-Bayesian generalization error bounds [7–9]. Moreover, a recent theoretical study rigorously connects the flatness of the loss surface to generalization in classification tasks under the assumption that labels are (approximately) locally constant [6]. Leveraging the great progress from the deep learning community, we aim to study the generalization properties of bio-plausible learning rules from a geometric perspective.

## 3  Results

In this section, we first describe the network and learning setup we use (Figure 1A). Next, we present a number of numerical experiments where we compute the generalization gap directly on three commonly used ML and neuroscience tasks (Figure 2A-C), for truncation-based bio-plausible gradient approximations. For the same experiments, we also quantify loss landscape curvature along learning trajectories, and connect these quantities to generalization behavior (Figures 2D-F and 3). Finally, we provide theoretical arguments and a theorem that explains how gradient alignment in bio-plausible gradient approximations can affect curvature preference (Theorem 1, Figure 3) and thus, generalization. Through additional experiments, we verify the predictive power of our theory. We conclude with discussions on potential remedies used by the brain (Appendix Figure 5) and future directions.

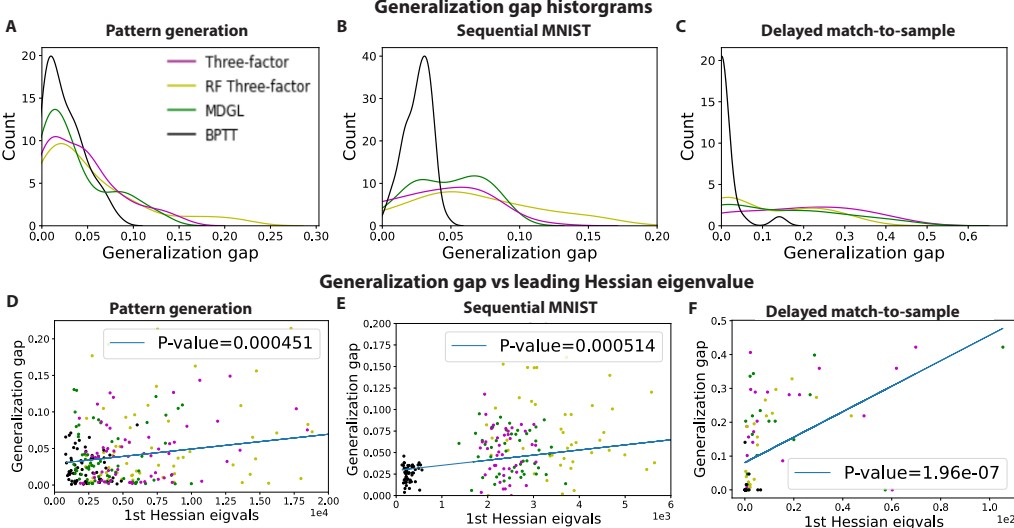

Figure 2: **Bio-plausible temporal credit assignment rules show worse and more variable generalization gap, which can be informed by loss landscape curvature**. A-C) Generalization gap distributions computed at the end of training across different random weight initializations for several well-known neuroscience and machine learning tasks. The higher the generalization gap, the worse the generalization performance. BPTT (black), bio-plausible alternatives (magenta, yellow and green). D-F) Scatter plots showing the trend of generalization gap v.s. leading loss Hessian eigenvalue across many runs; each point corresponds to a single run of the same runs as in A-C.

### 3.1  Network and learning setup

The detailed governing equations of our setup can be found in Methods (Appendix A). We consider a RNN with $N_{in}$ input units, $N$ hidden units and $N_{out}$ readout units (Figure 1A). We verified that trends hold for different network sizes and refer the reader to Appendix A.3 for more details. The

update formula for $h_t \in \mathbb{R}^N$ (the hidden state at time $t$) is governed by:

$$h_{t+1} = \phi(W_h f(h_t), W_x x_t), \tag{1}$$

where $\phi(\cdot) : \mathbb{R}^N \to \mathbb{R}^N$ is the hidden state update function, $f(\cdot) : \mathbb{R}^N \to \mathbb{R}^N$ is the activation function, $W_h \in \mathbb{R}^{N \times N}$ (resp. $W_x \in \mathbb{R}^{N_{in} \times N}$) is the recurrent (resp. input) weight matrix and $x \in \mathbb{R}^{N_{in}}$ is the input. For $\phi$, we consider a discrete-time implementation of a rate-based recurrent neural network (RNN) similar to the form in [136] (details in Appendix A). Readout $\hat{y} \in \mathbb{R}^{N_{out}}$, with readout weights $w \in \mathbb{R}^{N_{out} \times N}$, is defined as

$$\hat{y} = \langle w, f(h_t) \rangle. \tag{2}$$

We performed experiments on three tasks: sequential MNIST [137], pattern generation [138] and delayed match-to-sample tasks [139]. The objective is to minimize scalar loss $L \in \mathbb{R}$, which is defined as

$$L(W_h) = \begin{cases} \frac{1}{2TB} \sum_{i=1}^{B} \sum_{t=1}^{T} \sum_{k=1}^{N_{out}} (\hat{y}_{k,t}^{(i)} - y_{k,t}^{(i)})^2, & \text{for regression tasks} \\ \frac{-1}{TB} \sum_{i=1}^{B} \sum_{t=1}^{T} \sum_{k=1}^{N_{out}} \pi_{k,t}^{(i)} log \hat{\pi}_{k,t}^{(i)}, & \text{for classification tasks} \end{cases} \tag{3}$$

given target readout $y \in \mathbb{R}^{N_{out}}$, task duration $T \in \mathbb{R}$ and batch size $B \in \mathbb{R}$. $\pi_{k,t} \in \mathbb{R}$ is the one-hot encoded target for readout unit $k$ at time $t$ and $\hat{\pi}_{k,t} = \text{softmax}_k(\hat{y}_{1,t}, \ldots, \hat{y}_{N_{OUT},t}) = \exp(\hat{y}_{k,t})/\sum_{k'} \exp(\hat{y}_{k',t})$ is the predicted category probability.

Different learning algorithms examined in this work are BPTT (our benchmark), which update weights by computing the exact gradient ($\nabla L(W_h) \in \mathbb{R}^{N \times N}$):

$$\Delta W_h = -\eta \nabla L(W_h), \tag{4}$$

and three SoTA bio-plausible learning rules that update weights using approximate gradient:

$$\widehat{\Delta W_h} = -\eta \tilde{\nabla} L(W_h), \tag{5}$$

where $\tilde{\nabla} L(W_h) \in \mathbb{R}^{N \times N}$ denotes a gradient approximation and $\eta \in \mathbb{R}$ denotes the learning rate. These three learning rules are explained further in Appendix A (Methods) but we note that these bio-plausible learning rules are based on truncations of dependency paths — on the computational graph for the exact gradient— that are biologically implausible to compute (Figure 1A). In all figures, learning rules are labeled as "Three-factor" (symmetric e-prop), "RF Three-factor" (RFLO) and "MDGL", respectively. We remark that the focus here is on comparing artificial to bio-plausible learning rules, rather than between biological rules. Finally, we note that tasks were learned with mostly comparable training accuracies for all learning rules, and that generalization gaps reflect a testing departure from these values. We refer the reader to the Appendix for more details (Appendix A.3).

### 3.2 Generalization gap and loss landscape curvature

To study generalization performance, our first step is to compute the generalization gap empirically. Generalization gap is defined as train accuracy minus test accuracy; the larger the generalization gap, the worse the generalization performance. For various learning rules, we plot the generalization gap histogram at the end of training across runs with distinct initializations; different colors represent different learning rules (Figure 2A-C). Notice that these bio-plausible rules achieve worse and more variable generalization performance than their machine learning counterpart (BPTT).

We now investigate if the generalization gap behavior described above correlates well with Loss landscape curvature. We use the leading eigenvalue of the loss' Hessian (where derivatives are taken with respect to model parameters) as a measure for curvature following previous practice [121] and note that it is practical for both empirical [118] and theoretical analyses (Theorem 1). There exist other measures for flatness and due to the scale-dependence issue of Hessian spectrum [133], we also test using parameter scale-independent measures (see Appendix Figures 7 and 8). When we plot generalization gap points from Figure 2A-C against the corresponding leading Hessian eigenvalue, a statistically significant correlation is observed (Figure 2D-F). We also observed such correlation across runs with the learning rule fixed (Appendix Figure 12). We note that we did not expect this relationship to be very tight since in addition to the worse generalization gap on average, bio-plausible learning rules exhibit increased variance. This is an important and consistent trend we observed but might have been overlooked in previous studies.

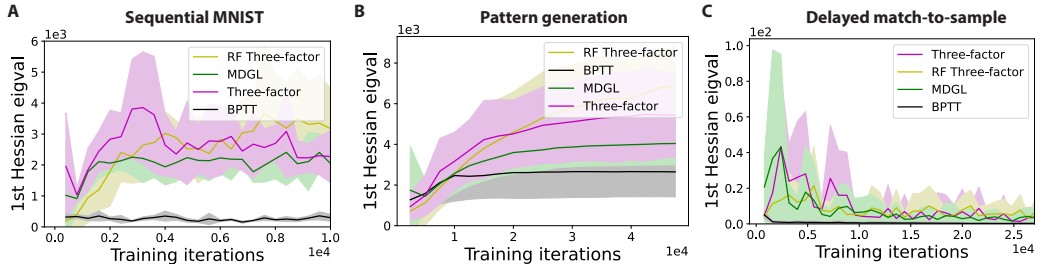

Figure 3: **Bio-plausible gradient approximations tend to approach high curvature regions in loss landscape**. Dominant Hessian eigenvalues are plotted throughout training for bio-plausible learning rules and BPTT. This analysis is done for A) sequential MNIST, B) pattern generation and C) delayed match-to-sample tasks. Solid lines/shaded regions: mean/standard deviation of dominant Hessian eigenvalue curves across five independent runs.

So far, we investigated the endpoints of optimization trajectories, where training performance has converged. Now, we visualize the whole training trajectory. This done is for two reasons: 1) account for early stopping that can halt training anywhere along the trajectory due to time constraints; 2) flatness during training could give indications of avoiding or escaping high-curvature regions. We observe that the biologically motivated gradient approximations tend to rapidly approach high-curvature regions compared to their machine learning counterpart (Figure 3). Together, these results demonstrate a clear trend and a link between the generalization gap and loss landscape curvature: both being increased and more variable for bio-plausible rules. We also stopped BPTT early to match the test accuracy of the three-factor rule, and observed similar trends as in the main text (Appendix Table 1). We also note that the curvature convergence behavior seems to be a shared problem of temporal truncations of the gradient (Appendix Figure 9), which is what the existing bio-plausible gradient approximations for RNNs are based on. Next, we provide a theoretical argument as to why truncated temporal credit assignment rules favor high curvature regions of the loss landscape.

### 3.3 Theoretical analysis: link between curvature and gradient approximation error

We discussed how generalization can be linked to curvature, and we now examine the link between curvature and gradient alignment during learning dynamics. We represent approximate gradients in a rule-agnostic manner, where an arbitrary approximation is represented in terms of its component along the gradient direction plus an arbitrary orthogonal vector (Figure 4A):

$$\vec{\tilde{g}} = \rho\vec{g} + \vec{e}, \tag{6}$$

where $\vec{g} := \nabla L(W_h) \in \mathbb{R}^{N^2}$ and $\vec{\tilde{g}} \in \mathbb{R}^{N^2}$ are the exact and approximate gradients, respectively (we reshaped $\nabla L(W_h)$ into a vector here); $\vec{e} \in \mathbb{R}^{N^2}$ is an arbitrary orthogonal vector to $\vec{g}$. Here, scalar $\rho \in \mathbb{R}$ represents the relative step length along the gradient direction that the approximate rule is making. As we will see in Theorem 1, $\rho$ is an important quantity in our analysis. One can easily compute $\rho$ from $\vec{g}$ and $\vec{\tilde{g}}$ by $\rho = \frac{\vec{\tilde{g}}^T \vec{g}}{\vec{g}^T \vec{g}}$.

We express weight updates as discrete dynamical systems (with weights $W_h$ as the state variables):

$$W_h^+ \leftarrow W_h^- + F(W_h^-) = W_h^- - \eta\nabla L(W_h^-), \text{ for BPTT} \tag{7}$$

$$W_h^+ \leftarrow W_h^- + \hat{F}(W_h^-) = W_h^- - \eta\tilde{\nabla}L(W_h^-), \text{ for an approximate rule,} \tag{8}$$

where $\eta$ is the learning rate, $\tilde{\nabla}$ is an approximate gradient, and $F : \mathbb{R}^{N^2} \rightarrow \mathbb{R}^{N^2}$ (resp. $\hat{F} : \mathbb{R}^{N^2} \rightarrow \mathbb{R}^{N^2}$) denotes the map defined by BPTT (resp. an approximate) weight update rule. Notation $W^+$ and $W^-$ denotes $W$ at the next and current step, respectively. We note in passing that dynamical systems view of weight updates have been used previously [140, 141].

We introduce additional notations before presenting Theorem 1. $J \in \mathbb{R}^{N^2 \times N^2}$ (resp. $\hat{J} \in \mathbb{R}^{N^2 \times N^2}$) is the Jacobian of the dynamical system of BPTT (resp. an approximate rule) in Eq. 7 (resp. Eq. 8). $\lambda_1^J \in \mathbb{R}$ (resp. $\widehat{\lambda_1^J} \in \mathbb{R}$) is the leading eigenvalue for BPTT (resp. an approximate rule) Jacobian.

$\lambda_1^H \in \mathbb{R}$ is the leading eigenvalue of the loss' Hessian matrix. $W_B^* \in \mathbb{R}^{N \times N}$ (resp. $W_e^* \in \mathbb{R}^{N \times N}$) is the final fixed point for BPTT (resp. an approximate rule). We now present Theorem 1.

**Theorem 1.** *Consider an RNN defined in Eq. 1 with a single scalar output $\hat{y}$ and least squares loss as in Eq. 3 presented only at the last time step $T$, and weights are updated according to the difference equation for BPTT 7 (resp. an approximate rule 8) on a single example (stochastic gradient descent) using learning rate $\eta_B$ (resp. $\eta_e$). In the limit of stable fixed point convergence with zero training error, the dominant loss' Hessian eigenvalue attained by BPTT (resp. approximate rule) is bounded by $|\lambda_1^H(W_B^*)| < \frac{1}{\eta_B}$ (resp. $|\lambda_1^H(W_e^*)| < \frac{1}{|\rho|\eta_e}$).*

*Proof.* Full proof is in Appendix B. Here is a summary of the main steps involved:

1. The Jacobian of BPTT dynamical system (Eq. 7) is the loss' Hessian scaled by a constant;

2. Using the above relationship, we can bound $|\lambda_1^H|$ from $|\lambda_1^J| < 1$, which is the condition for a discrete-time dynamical system to converge to a fixed point

3. However, the link between the Jacobian of an approximate update rule and the loss' Hessian is less obvious. Thus, we derive a link between $|\widehat{\lambda_1^J}|$ and $|\lambda_1^J|$, and then apply the step above

$\square$

The consequence of the upper bound derived in Theorem 1 is that truncated gradient rules can converge to minima with a higher dominant Hessian eigenvalue than BPTT, with the leading eigenvalue bound inversely proportional to $|\rho|$ ($|\rho| < 1$ usually). In practice, $\rho$ can vary depending on task settings and in our setup, we observed it to be somewhere between 0.02 and 0.3. We remark that this higher upper bound is consistent with the increased spread of curvature and generalization observed for bio-plausible rules in experiments. Theorem 1 highlights scalar $\rho$ (Eq. 6), relative step length along the gradient direction, as an important factor in the curvature bound. To test that, we eliminate the factor of $\rho$ by reducing the learning rule of BPTT such that its step length is matched to that of the three-factor rule. This resulted in the blue curves in Figure 4, which is still trained using BPTT but with the update scaled by a factor of $\rho$. By matching the step length of BPTT and a three-factor rule along the gradient direction, similar convergence behaviors were observed. Similar observations were also made when the matching step experiment was repeated at three times the learning rate for all rules (Appendix Figure 12C). **This result then attributes the curvature preference behavior to relative step length along the gradient direction, and thereby indicating a link between curvature and gradient alignment under certain conditions**. Consistent with earlier results, when the step length of BPTT is matched to that of a three-factor rule, its generalization performance also worsened (Appendix Figure 6).

We make a few more remarks regarding Theorem 1. $\rho$ is usually less than 1 because otherwise the additional orthogonal component $\vec{e}$ would imply a larger update size for the approximate rule compared to BPTT. This would make the approximate rule more prone to numerical instabilities (Eq. 9) and would require a change in solver hyperparameters such as overall learning rate which in turn, influence the update size. In our experiments, when the three-factor rule was scaled up to match the along-gradient update sizes of BPTT, we quickly ran into numerical overflow (values of NaNs in the network), which is expected for a very large learning rate. The exact point for when this numerical instability is reached depends on many factors such as the model, task, numerical precision as well as the consistency of update direction. We have also equated numerical instabilities in simulations as a proxy for situations problematic for the brain in Discussion. Curiously, $\vec{e}$ did not seem to play a helpful role in finding flatter minima; this could be due to that approximation error $\vec{e}$ is not well aligned with the sharpest directions (Appendix Figure 10). In fact, $\vec{e}$ being orthogonal to the leading Hessian eigenvectors is a consequence of the assumptions behind Theorem 1. Despite making assumptions including scalar output, MSE loss and loss available at the last step, our empirical results indicate that our conclusions also extend to other setups: vector output, cross-entropy loss and loss that accumulates over time steps (Figure 4). In Appendix B.3, we discuss the generality of Theorem 1 by using quadratic expansion of the loss function and assuming that $\vec{e}$ is orthogonal to the leading Hessian eigenvectors.

One might assume increasing the learning rate of the approximate rules could compensate for the reduced step length due to $\rho$. However, this can raise other issues. Suppose $\Delta W = -\eta\vec{g}$ for the

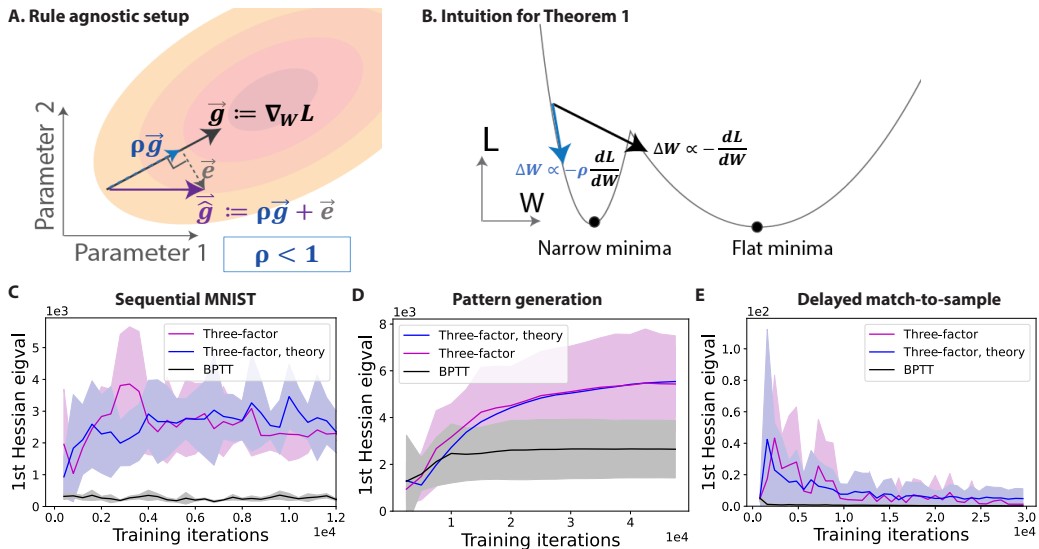

Figure 4: **Preference for high curvature regions connected to worse gradient alignment under certain conditions**. A) An arbitrary gradient approximation, $\hat{\vec{g}}$, its component along the gradient direction, $\rho\vec{g}$, and orthogonal to the gradient $\vec{e}$ (Eq. 6). The scalar $\rho$ represents the relative step length along the gradient direction. B) Illustration for Theorem 1: if gradient $\vec{g}$ is aligned with the sharpest directions but error $\vec{e}$ is not (see Appendix Figure 10), smaller step length along the gradient direction can make it harder to step over narrow minima. C-E) Leading loss Hessian eigenvalue v.s. training iteration for the three-factor rule, BPTT, and modified BPTT (three-factor, theory). For the latter, step length along the gradient direction of BPTT was matched to three-factor rule by multiplying BPTT update with a factor of $\rho$, which recovers curvature trends of the three-factor rule.

exact gradient and we increase the learning rate of the approximate update until $\widehat{\Delta W} = -\eta\vec{g} - \eta\vec{e}$ (magnitude of $\vec{e}$ is scaled accordingly), and because $\vec{g} \perp \vec{e}$:

$$\|\widehat{\Delta W}\|^2 = \eta^2\|\vec{g}\|^2 + \eta^2\|\vec{e}\|^2 > \eta^2\|g\|^2 = \|\Delta W\|^2 \tag{9}$$

In other words, if approximate updates $\widehat{\Delta W}$ make the same amount of progress as BPTT along the gradient direction, $\widehat{\Delta W}$ would have a larger magnitude due to the orthogonal component $\vec{e}$, and large update magnitude can be very problematic for numerical stability [142]. Thus, the magnitude of $\vec{e}$ limits the learning rate that can be used. **Because of the numerical issues associated with increasing the learning rate for approximate rules (due to $\vec{e}$), the differences in generalization and curvature convergence between rules cannot be reduced by increasing the learning rate for approximate rules.** To balance between this numerical stability issue and the potential benefit of large learning rates, one can consider using a large learning rate early in training to prevent premature stabilization in sharp minima followed by gradual decay to mitigate the stability issue (see Appendix Figure 5).

Taken together, Theorem 1 and Eq. 9 connect gradient alignment with the curvature of the converged solution under certain conditions. **For an approximation that is aligned poorly with the exact gradient, large orthogonal approximation error vector $\vec{e}$ limit the step length for numerical stability reasons (Eq. 9) and small relative step length $\rho$ correspond to a larger curvature bound (Theorem 1).**

## 4   Conclusion

While developing bio-plausible learning rules is of interest for both answering neuroscience questions and searching for more efficient training strategies for artificial networks, the generalization properties of solutions found by these rules are severely underexamined. Through various well-known machine learning and working memory tasks, we first demonstrate empirically that existing bio-plausible

temporal credit assignment rules attain worse generalization performance, which is consistent with their tendency to converge to high-curvature regions in loss landscape. Second, our theoretical analysis offers an explanation for this preference for high curvature regions based on worse alignment to the true gradient. This regime corresponds to the situation where the step length along the gradient direction is small and the approximation error vector is large. Finally, we test this theory empirically by matching the relative step length along the gradient direction resulting in similar convergence behavior (Figure 4).

## 5  Discussion

Our study — a stepping stone toward understanding biological generalization using deep learning methods — raises many exciting questions for future investigations, both on the front of stronger deep learning theory and more sophisticated biological ingredients.

**Deep learning theory and its implications:**  In this study, we investigate generalization properties using loss landscape flatness, a promising predictor with recent rigorous connection to generalization gap [6]. Yet, to what extent can flatness explain generalization is still an open question in deep learning. For instance, curvature is a local measure, which means that its informativeness of robustness against global perturbation is limited. Moreover, the theoretical association between flatness and generalization is provided in the form of upper bound [6, 9] and the bound may not always be tight, which is consistent with more variability in the generalization gap for bio-plausible learning rules but offers less predictive power. Despite observing a significant correlation between the generalization gap and a curvature-based measure in Figure 2, the relationship appears to be messy, suggesting other factors involved in explaining generalization. Given that developing better predictors of generalization is still a work in progress [4], we anticipate stronger theoretical tools to be applied for studying biological generalization in the future. Our results are also consistent with existing findings linking learning rate to loss landscape curvature in deep networks [123]. In the case of bio-plausible gradient approximations, small step length along the gradient direction cannot be compensated by increasing the learning rate, as that would inadvertently increase the error vector, causing numerical issues (Eq. 9). Curiously, the approximation error vector $\vec{e}$ did not seem to play a role other than restricting the learning rate. While it is well-known that stochastic gradient noise (SGN) can help with finding flat minima due to the alignment of SGN covariance and Hessian near minima [122, 143–145], that may not apply to approximation error vector $\vec{e}$ resulting primarily from temporal truncation of the gradient (see Figure 1A and Appendix Figure 10). This indicates that noise with different properties (e.g. different directions) could affect generalization differently, thereby motivating future investigations into how a broad range of biological noises — which may differ from noise in ML optimization (e.g. SGN) — can impact generalization.

Moreover, our results are closely related to a series of studies that examined the "catapulting" behavior in learning [146–150]. This can happen when the second order Taylor term of the loss function would dominate over the first, which would cause learning to cross the threshold for step size stability and "catapult" into a flatter region that can accommodate the step size. If the truncation noise is only aligned with the eigendirections associated with negligible eigenvalues, then it can only have limited contributions to the second-order Taylor term. On top of that, the orthogonal noise term would require a smaller learning rate to be used to avoid numerical issues, as explained earlier. Overall, these would lead to a weaker second-order Taylor term relative to the first for bio-plausible temporal credit assignment rules, which would then increase the threshold for step size stability. This increased stability is closely tied to the greater dynamical stability (for the weight update difference equation) of approximation rules predicted by Theorem 1, due to the correspondence between loss' Hessian matrix and the Jacobian matrix of the weight update difference equation (the correspondence is explained in Theorem 1 proof in Appendix B).

**Toward more detailed biological mechanisms:**  On the front of more sophisticated biological ingredients that may improve generalization performance, we see two lines of approaches: **1-** develop bio-plausible learning rules that align better with the gradient, as suggested by our rule-agnostic analysis (Theorem 1, Figure 4); and **2-** instead of studying learning rules in isolation, consider also other neural circuit components [151–158] that could interact with the learning rule. An important component would be the architecture, including connectivity structure and neuron model, found through evolution [156] (see also [159]). To address our main question, this study varies learning rules

while holding data, objective function and architecture constant (see [10]). However, these different components can interact, and more sophisticated architecture can facilitate task learning [159–166]. Given the exploding parameter space resulting from such interactions, we believe it requires careful future analysis and is outside of the scope for this one paper.

Additionally, learning rate modulation [167, 168] could be one of many possible remedies employed by the brain. We conjecture that neuromodulatory mechanisms could be coupled with these learning rules to improve the convergence behavior through our scheduled learning rate experiments (Appendix Figure 5), where an initial high learning rate could prevent the learning trajectory from settling in sharp minima prematurely followed by gradual decay to avoid instabilities. One possible way to realize such learning rate modulation could be through serotonin neurons via uncertainty tracking, where the learning rate is high when the reward prediction error is high (this can happen at the beginning of learning) [169]. Since the authors of [169] showed that inhibiting serotonin led to failure in learning rate modulation, we conjecture that such inhibition might have an impact on the generalization performance of learning outcomes. On the topic of balancing numerical instabilities and potential advantages of large learning rates, while the analog nature of biology may seem to avoid finite precision representation in digital computers that give rise to numerical instabilities, the same problems that lead to numerical instabilities in digital computers, such as big ranges between quantities added or multiplied, remains an issue for biology since quantities must be stored in noisy activity patterns of neurotransmitter release. Future investigations could investigate potential homeostatic mechanisms that regulate biological quantities to avoid such instabilities, thereby enabling larger "learning rates" to be used so as to find flatter minima. Other ingredients for future investigations could include intrinsic noise with certain structures [153, 170, 171] (including directions and bias/variance properties) that would make them more favorable for generalization. Taken together, we hope to see follow-up investigations — riding on the rapid advancements both at the front of deep learning theory and sophisticated biological mechanisms — into how the brain attains solutions that generalize.

# 6   Acknowledgement

We thank Gauthier Gidel, Jonathan Cornford, Aristide Baratin, Thomas George and Mohammad Pezeshki for insightful discussions and helpful feedback at an early stage of this project. We also thank Henning Petzka and Michael Kamp for helpful email exchanges. This research was supported by NSERC PGS-D (Y.H.L.); NSF AccelNet IN-BIC program, Grant No. OISE-2019976 AM02 (Y.H.L.); Vanier Canada Graduate scholarship (A.G.); Healthy Brains for Healthy Lives fellowship (A.G.); CIFAR Learning in Machines and Brains Program (B.A.R.), NSERC Discovery Grant RGPIN-2018-04821 (G.L), FRQS Research Scholar Award, Junior 1 LAJGU0401-253188 (G.L.), Canada Research Chair in Neural Computations and Interfacing (G.L.), Canada CIFAR AI Chair program (G.L. & B.A.R.). We also thank the Allen Institute founder, Paul G. Allen, for his vision, encouragement, and support.

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
