# A Methods

## A.1 Hessian eigenspectrum analysis

As mentioned, we focus on the leading Hessian eigenvalue because it has been used previously [121, 172] and is very feasible for both empirical [118, 173] and theoretical analyses (Theorem 1). The leading Hessian eigenvalue can be computed by performing power iterations on the Hessian vector product without knowing the full Hessian matrix (Algorithm 2 in [118]). To find multiple top eigenvalues, e.g. top 200 eigenvalues, one can use the generalized power method via QR decomposition [174]. We focused on the loss' Hessian for recurrent weights but observed a similar trend for input weights as well. We set the tolerance for stopping to $1e - 6$.

Due to the scale-dependence issue of Hessian spectrum [133], we also used scale-independent measures. For instance, we examined the power-law decay coefficient for the Hessian eigenvalues (Figure 7 in Appendix). We also looked at the recently proposed relative flatness measure [6] (Figure 8 in Appendix). We used the code in [175] to fit a power-law distribution to the top 200 eigenvalues. We found similar results had we chosen the top 50 or 100 eigenvalues instead, and 200 was chosen mainly due to computational load. We note that the link of power-law decay to generalization has also been examined in some recent studies [120, 176, 177].

## A.2 Network setup and learning rule implementations

**Neuron Model**: We consider a discrete-time implementation of a rate-based recurrent neural network (RNN) similar to the form in [136]. The model denotes the internal hidden state as $h_t$ and the observable states, i.e. firing rates, as $f(h_t)$ at time $t$, and we use ReLU activation for $f$. The dynamics of those states are governed by

$$h_{t+1} = \alpha h_t + (1 - \alpha) \left( W_h f(h_t) + W_x x_t \right), \tag{10}$$

where $\alpha = e^{-dt/\tau_m}$ denotes the leak factor for simulation time step $dt$ and membrane time constant $\tau_m$, $W_h$ denotes the weight of the recurrent synaptic connection, $W_x$ denotes the strength of the input synaptic connection and $x_t$ denotes the external input at time $t$. we use subscripts to represent indices of neurons and time steps. For instance, $h_{i,t}$ represents the hidden activity $h$ of neuron $i$ at time $t$. $W_{h,ij}$ represents the $(ij)^{th}$ entry of recurrent weight matrix $W_h$. Model in Eq. 10 was used for the sequential MNIST and pattern generation tasks.

We mention in passing that the choice of ReLU activation, which has a discontinuous first derivative, means that the loss Hessian matrix is not guaranteed to be symmetric. A real matrix that is not symmetric can have complex eigenvalues come in conjugate pairs, and if they were amongst the top eigenvalues, power iterations may not converge. However, all iterations have converged in our experiments as mentioned above. Also, because of potential technical issues resulting from non-symmetric Hessian matrices, we foresee challenges in applying our methodology to spiking neural networks (SNNs), which have discontinuous activation functions. Due to the energy efficiency and biological realism of SNNs [96, 178–187], we believe extending to SNNs is an important future direction.

For the delayed match to sample task, which is a working memory task, it was found in [13] and [11] that units with an adaptive threshold as an additional hidden variable can play an important role in the computing capabilities of RNNs. Thus, we implemented adaptive threshold neuron units [164] for that task. In our rate-based implementation, this turns out to be a simple addition of a second hidden variable $b_t$ that represents the dynamic threshold component:

$$\begin{aligned} h_{t+1} &= \alpha h_t + (1 - \alpha) \left( W_h f(h_t - b_t) + W_x x_t \right), \\ b_{t+1} &= \beta b_t + (1 - \beta) f(h_t - b_t), \end{aligned} \tag{11}$$

where $b_{j,t}$ denotes the dynamic threshold that adapts based on past neuron activity. The decay factor $\beta$ is given by $e^{-dt/\tau_b}$ for simulation time step $dt$ and adaptation time constant $\tau_b$, which is typically chosen on the behavioral task time scale [13].

**Network output and loss function**:

Readout $\hat{y}$ is defined as

$$\hat{y} = \langle w, f(h_t) \rangle \tag{12}$$

for readout weights $w$.

We quantify how well the network output $\hat{y}$ matches the desired target $y$ using loss function $L$, which is defined as

$$L(W_h) = \begin{cases} \frac{1}{2TB} \sum_{i=1}^{B} \sum_{t=1}^{T} \sum_{k=1}^{N_{out}} (\hat{y}_{k,t}^{(i)} - y_{k,t}^{(i)})^2, & \text{for regression tasks} \\ \frac{-1}{TB} \sum_{i=1}^{B} \sum_{t=1}^{T} \sum_{k=1}^{N_{out}} \pi_{k,t}^{(i)} log \hat{\pi}_{k,t}^{(i)}, & \text{for classification tasks} \end{cases} \quad (13)$$

for target output $y$, task duration $T$, $N_{out}$ output neurons and batch size $B$. $\pi_{k,t}$ is the one-hot encoded target and $\hat{\pi}_{k,t} = \text{softmax}_k(\hat{y}_{1,t}, \ldots, \hat{y}_{N_{OUT},t}) = \exp(\hat{y}_{k,t}) / \sum_{k'} \exp(\hat{y}_{k',t})$ is the predicted category probability.

**Biological gradient approximations (truncation-based)**

The goal of this subsection is to explain where the approximation happens for each of the bio-plausible learning rules. For full details regarding these rules, we encourage the reader to refer to the respective references. We start by writing down the gradient in terms of real-time recurrent learning (RTRL) factorization:

$$\frac{\partial L}{\partial W_{h,ij}} = \sum_{l,t} \frac{\partial L}{\partial h_{l,t}} \frac{\partial h_{l,t}}{\partial W_{h,ij}}, \quad (14)$$

Key problems that RTRL poses to biological plausibility and computational cost reside in the second factor $\frac{\partial h_{l,t}}{\partial W_{h,ij}}$ that arises during the factorization of the gradient (Eq. 14). The factor $\frac{\partial h_{l,t}}{\partial W_{h,ij}}$ keeps track of all recursive dependencies of $h_{l,t}$ on weight $W_{h,ij}$ arising from recurrent connections. These recurrent dependencies can be obtained recursively as follows:

$$\frac{\partial h_{l,t}}{\partial W_{h,ij}} = \frac{\partial h_{j,t}}{\partial W_{h,ij}} + \sum_m \frac{\partial h_{l,t}}{\partial h_{m,t-1}} \frac{\partial h_{m,t-1}}{\partial W_{h,ij}}$$

$$= \frac{\partial h_{l,t}}{\partial W_{h,ij}} + \frac{\partial h_{l,t}}{\partial h_{l,t-1}} \frac{\partial h_{l,t-1}}{\partial W_{h,ij}} + \underbrace{\sum_{m \neq l} W_{h,lm} f'(h_{m,t-1}) \frac{\partial h_{m,t-1}}{\partial W_{h,ij}}}_{\text{depends on all weights } W_{h,lm}}. \quad (15)$$

Thus, the factor $\frac{\partial h_{l,t}}{\partial W_{h,ij}}$ poses a serious problem for biological plausibility: it involves **nonlocal** terms that should be inaccessible to neural circuits, i.e. that knowledge of all other weights in the network is required in order to update the weight $W_{h,ij}$.

**RFLO** [12] (labeled as "RF Three-factor") and **symmetric e-prop** [11] (labeled as "Three-factor") seek to address this by truncating the expensive nonlocal terms in Eq. 15 so that the updates to weight $W_{h,ij}$ would only depend on pre- and post-synaptic activity:

$$\widehat{\frac{\partial h_{l,t}}{\partial W_{h,ij}}} = \begin{cases} \frac{\partial h_{i,t}}{\partial W_{h,ij}} + \frac{\partial h_{i,t}}{\partial h_{i,t-1}} \widehat{\frac{\partial h_{i,t-1}}{\partial W_{h,ij}}}, & l = i \\ 0, & l \neq i \end{cases} \quad (16)$$

which results in a much simpler factor than the triple tensor in Eq. 15.

After the truncation, RFLO and e-prop implement:

$$\widehat{\frac{\partial L}{\partial W_{h,ij}}} = \sum_t \frac{\partial L}{\partial h_{i,t}} \widehat{\frac{\partial h_{i,t}}{\partial W_{h,ij}}}, \quad (17)$$

$$\widehat{\frac{\partial h_{i,t}}{\partial W_{h,ij}}} = \frac{\partial h_{i,t}}{\partial W_{h,ij}} + \frac{\partial h_{i,t}}{\partial h_{i,t-1}} \widehat{\frac{\partial h_{i,t-1}}{\partial W_{h,ij}}}. \quad (18)$$

The main difference between symmetric e-prop and RFLO implementation is that symmetric feedback is used for symmetric e-prop, i.e. output weight $w$ is used as the feedback weight for the $\frac{\partial E}{\partial h}$, whereas RFLO uses fixed random feedback weights [75] for greater biological plausibility. We note in passing that the authors of e-prop have tested their formulation with fixed random feedback weights as well. **MDGL** [13] also truncates RTRL, but it restores some of the non-local dependencies – those within one connection step — that could potentially be communicated via mechanisms similar

to the abundant cell-type-specific local modulatory signaling unveiled by recent transcriptomics data [89, 90]. With that, the expensive memory trace term in Eq. 15 becomes

$$\frac{\partial h_{l,t}}{\partial W_{h,ij}} \approx \begin{cases} W_{h,li} f'(h_{i,t-1}) \widehat{\frac{\partial h_{i,t-1}}{\partial W_{h,ij}}}, & i \neq l \\ \frac{\partial h_{i,t}}{\partial W_{h,ij}} + \frac{\partial h_{i,t}}{\partial h_{i,t-1}} \widehat{\frac{\partial h_{i,t-1}}{\partial w_{ij}}}, & i = l \end{cases}$$

(19)

MDGL involves one additional approximation: replace $W_{h,li}$ with type-specific weights $W_{ab}$ to mimic the cell-type-specific nature of local modulatory signaling (for cell $i$ in group $a$ and cell $j$ in group $b$, where $a, b \in C$ for a total of $C$ cell groups). For simplicity, we just used $W_{ab} = W_{li}$, i.e. without cell-type approximation. This results in overall MDGL implementation as

$$\widehat{\frac{\partial L}{\partial W_{h,ij}}} = \sum_t \frac{\partial L}{\partial h_{i,t}} \widehat{\frac{\partial h_{i,t}}{\partial W_{h,ij}}} + \widehat{\frac{\partial h_{i,t-1}}{\partial W_{h,ij}}} \sum_l W_{h,li} \frac{\partial L}{\partial h_{l,t}},$$

(20)

Interpretation of the above update rule in terms of biological processes can be found in the MDGL paper [13, 188].

We note that input, recurrent and output weights were all being trained. This section illustrates the approximate gradient for updating recurrent weights $W_h$, and similar expressions were obtained for updating input weights $W_x$. The approximations, however, did not apply to output weights, as the gradient for that would not violate the aforementioned issue of nonlocality (Eq. 15).

### A.3 Simulation details

We used TensorFlow [189] version 1.14 and based it on top of [190]. We modified the code for rate-based neurons (Eq. 10 and 11). [1] We used the code in [175] for the power-law analysis (Figure 7 in Appendix). SGD optimizer was used to study the effect of gradient approximation in isolation without the complication of additional factors, as Adam optimizer with adaptive learning rate could convolute our matching step length experiments in Figure 4. That said, we verified that the curvature convergence behavior is also observed for Adam optimizer (Figure 11 in Appendix. Learning rates were optimized by picking within $\{3e-4, 1e-3, 3e-3, 1e-2, 3e-2, 1e-1\}$ for each algorithm. For the sequential MNIST task, we explored batch sizes within $\{64, 256, 1024\}$. For the sequential MNIST task, these hyperparameters were optimized based on validation performance (the validation set loaded using $tensorflow.examples.tutorials.mnist$). For the two other tasks, these hyperparameters were optimized based on the training performance, but we also tried optimizing on the test set and observed similar trends. Trainings were stopped when both the loss and leading Hessian eigenvalue stabilized. As stated, we repeated runs with different random initialization to quantify uncertainty and weights were initialized similarly as in [12].

Simulations were completed on a computer server with x2 20-core Intel(R) Xeon(R) CPU E5-2698 v4 at 2.20GHz. The average time to complete one run of sequential MNIST, pattern generation and delayed match to sample tasks in Figure 3 were approximately 2 hours, 1 hour and 1 hour, respectively. Since the computation of second order gradient becomes prohibitively expensive as sequence length $T$ becomes large, all tasks involved no more than 50 time steps. For instance, this was achieved for the sequential MNIST task using the row-by-row implementation. Using fewer time steps, however, should not affect the general trend as the gradient truncation effects were still significant. Because of the use of fewer steps, we dropped the leak factor $\alpha$ in Eq. 10 (i.e. set $\alpha = 0$).

For the matching step length experiments (Figure 4), we simply obtained $\rho = \frac{\vec{g}^T \vec{g}}{\vec{g}^T \vec{g}}$ for the three-factor learning rule and scaled BPTT updates by that amount. For scheduled learning rate experiments (Figure 5), the additional hyperparameters included initial learning rate, decay percentage and decay frequency. We used an initial learning rate that was three times the uniform rate (used in other figures) and decay the learning rate by $80\%$ every X iterations, where X was roughly the total number of training iterations (used in other figures) divided by 30. Since the point of that figure was to show that learning rate scheduling could lead to flatter minima than using a fixed learning rate, we did not search extensively across these additional hyperparameters as the first set of hyperparameters we tried was enough to demonstrate that point.

---

[1]Our code is available: `https://github.com/Helena-Yuhan-Liu/BiolHessRNN`.

For the pattern generation task, our network consisted of $N = 30$ neurons described in Eq. 10. Input to this network was provided by a random Gaussian input ($N_{in} = 1$). The fixed target signal had a duration of 50 steps and was given by the sum of four sinusoids, with a fixed period of 10, 40, 70 and 100 steps. For learning, we used the mean squared loss function. Training for this task used full batch. For testing, we perturbed the input with additive zero-mean Gaussian noise (with $\sigma$ picked uniformly between 0 and 0.2 across runs), to mimic the situation where the agent had to faithfully produce the desired pattern even under perturbations. Unlike the other tasks, this task measures accuracy by mean squared error, for which the lower the better. To maintain the convention of a higher generalization gap being worse, the generalization gap for this task was computed by test error minus train error.

For the delayed match to sample task, our network consisted of $N = 100$ neurons, which include 50 neurons with (Eq. 11) and 50 neurons without (Eq. 10) threshold adaptation. The task involved the presentation of two sequential cues, each taking on a binary value, lasting 2 steps and separated by a delay of 16 steps. Input to this network was provided by $N_{in} = 2$ neurons. The first (resp. second) input neuron sent a value of 1 when the presented cue took on a value of 1 (resp. cue 0), and 0 otherwise. The network was trained to output 1 (resp. 0) when the two cues have matching (resp. non-matching) values. For learning, we used the cross-entropy loss function and the target corresponding to the correct output was given at the end of the trial. Training for this task used full batch. For testing, we tested on increased delay, with the period picked uniformly between the training delay period and twice the training delay period, to mimic situations where the animal has to hold the memory longer than it did during the learning phase.

For the sequential MNIST task [137], our network consisted of $N = 128$ neurons described in Eq. 10. Input to this network was provided by $N_{in} = 28$ units that represented the grey-scaled value of a single row, totaling 28 steps and the network prediction was made at the last step. For learning, we used the cross-entropy loss function and the target corresponding to the correct output was given at the end of the trial. For testing, we used the existing MNIST test set [137] with additive zero-mean Gaussian input noise. As mentioned, this task did not train with a full batch but we found the trend to hold across different batch sizes.

Finally, we note that comparisons between BPTT and approximate rules were done at comparable training accuracies for the pattern generation and delayed match to sample tasks. For the sequential MNIST task, the three-factor rule achieved only around 70% training accuracy, but the training accuracy did not explain the curvature convergence behavior. To see this, while three-factor theory (blue in Fig 4), which corresponds to BPTT with reduced step length, achieves an accuracy of $> 95\%$ but still attains similar curvature convergence to that of the three-factor rule.

# B Theorem 1

## B.1 Proof of Theorem 1

*Proof.* First, we note that the Jacobian of the dynamical system for BPTT update (Eq. 7) is simply the loss Hessian scaled by $-\eta$. This implies that

$$|\lambda_1^J| = \eta_B |\lambda_1^H|, \tag{21}$$

where we remind the reader that $\lambda_1^J \in \mathbb{R}$ (resp. $\widehat{\lambda_1^J} \in \mathbb{R}$) is the leading eigenvalue for BPTT (resp. an approximate rule) Jacobian; $\lambda_1^H \in \mathbb{R}$ is the leading eigenvalue of the loss' Hessian matrix.

Second, with the assumption of a single output $\hat{y}$, loss presented only at the last time step $T$ and stochastic gradient descent (updates on a single data example as opposed to batch updates), least squares loss in Eq. 3 can be simplified to:

$$L(W_h) = \frac{1}{2}(\hat{y} - y)^2, \tag{22}$$

resulting in the following difference equations for discrete dynamical systems defined by BPTT (Eq. 7) and an approximate rule (Eq. 8):

$$\Delta W_h|_{\text{BPTT}} = F(W_h) = -\eta_B(\hat{y} - y)\nabla\hat{y}, \tag{23}$$

$$\Delta W_h|_{\text{approximate}} = \hat{F}(W_h) = -\eta_e(\hat{y} - y)\tilde{\nabla}\hat{y}, \tag{24}$$

where we remind the reader that $\tilde{\nabla}$ is the notation for an approximate gradient.

We then compute the Jacobian of the difference equations above:

$$J = -\eta_B \left( \nabla\hat{y}\nabla\hat{y}^T + (\hat{y} - y)\nabla^2\hat{y} \right) \quad \text{(for BPTT)} \tag{25}$$

$$\hat{J} = -\eta_e \left( \nabla\hat{y}\tilde{\nabla}\hat{y}^T + (\hat{y} - y)\nabla\tilde{\nabla}\hat{y} \right) \quad \text{(for an approximate rule).} \tag{26}$$

In the limit of zero error $(y - \hat{y}) = 0$, i.e. close to an optimum, the term involving $(y - \hat{y})$ becomes negligible. That simplifies the Jacobian to

$$J \approx -\eta_B \nabla\hat{y}\nabla\hat{y}^T$$
$$\hat{J} \approx -\eta_e \nabla\hat{y}\tilde{\nabla}\hat{y}^T. \tag{27}$$

In this case, $J$ and $\hat{J}$ are **rank-1** matrices. A rank-1 square matrix has only one nonzero eigenvalue, and by inspection, that one eigenvalue is

$$\text{For } J : |\lambda_1^J(W_B^*)| \quad = \eta_B \nabla\hat{y}^T\nabla\hat{y}\big|_{W_B^*} \tag{28}$$

$$\rightarrow |\lambda^H| \overset{(21)}{=} |\lambda_1^J|/\eta_B = \nabla\hat{y}^T\nabla\hat{y} \tag{29}$$

$$\text{For } \hat{J} : \quad |\widehat{\lambda_1^J}(W_e^*)| = \eta_e |\tilde{\nabla}\hat{y}^T\nabla\hat{y}|\Big|_{W_e^*}$$

$$\overset{(a)}{=} |\eta_e \rho \nabla\hat{y}^T\nabla\hat{y}|\big|_{W_e^*}$$

$$\overset{(29)}{=} |\rho\eta_e\lambda_1^H(W_e^*)|, \tag{30}$$

where equality (a) is explained as follows. We first remind the reader that $\rho$ is defined such that $\vec{\tilde{g}} = \rho\vec{g} + \vec{e}$ (Eq. 6). For the case of a scalar output $\hat{y}$, $\vec{\tilde{g}} = \frac{\partial L}{\partial\hat{y}}\tilde{\nabla}\hat{y}$ and $\vec{g} = \frac{\partial L}{\partial\hat{y}}\nabla\hat{y}$. So if we divide both sides of $\vec{\tilde{g}} = \rho\vec{g} + \vec{e}$ by $\frac{\partial L}{\partial\hat{y}}$ we get $\tilde{\nabla}\hat{y} = \rho\nabla\hat{y} + \vec{e}/\frac{\partial L}{\partial\hat{y}}$. Since we have $\vec{e} \perp \vec{g}$ by definition, then $\vec{e}^\top\nabla\hat{y} = 0$ because $\vec{g}$ is just a scaled $\nabla\hat{y}$ when the output is a scalar. This leads to $\tilde{\nabla}\hat{y}^T\nabla\hat{y} = (\rho\nabla\hat{y} + \vec{e}/\frac{\partial L}{\partial\hat{y}})^\top\nabla\hat{y} = \rho\nabla\hat{y}^T\nabla\hat{y}$.

Since we assume the gradient descent dynamical system converges to an optimum, this corresponds to an asymptotic stable fixed point. Hence, $|\lambda_1^J| < 1$ and $|\widehat{\lambda_1^J}| < 1$, which implies:

$$|\lambda_1^J(W_B^*)| < 1 \overset{(21)}{\rightarrow} \eta_B|\lambda_1^H(W_B^*)| < 1 \rightarrow |\lambda_1^H(W_B^*)| < \frac{1}{\eta_B} \tag{31}$$

$$|\widehat{\lambda_1^J}(W_e^*)| < 1 \overset{(30)}{\rightarrow} |\rho\eta_e\lambda_1^H(W_e^*)| < 1 \rightarrow |\lambda_1^H(W_e^*)| < \frac{1}{|\rho|\eta_e}. \tag{32}$$

$\square$

## B.2 Discussion on tightness of the bound

Following the derivation, it is clear that the tightness of the bound will depend on how close the magnitude of leading Jacobian eigenvalue is to 1 upon convergence. That is related to the distribution of minima flatness along the loss landscape, which impacts the probability of a rule converging to a minima with flatness in a certain range. Such distribution is likely problem dependent. If the loss were convex, there would just be one minimum and the question of minima preference would become irrelevant.

## B.3 Discussion on generality of Theorem 1

The proof above examines a special case where the Jacobian of weight update equations becomes rank-1. We remind the reader that for the case of multivariate loss, higher rank cases or batch updates, we would not have arrived at the rank-1 Jacobian step (Eq. 27). For many tasks considered in neuroscience, the rank 1 case can apply, which explains the validity of Theorem 1. We also note in passing that for the case of stochastic gradient descent, the Hessian Jacobian correspondence (Eq. 21) would point to loss' Hessian matrix evaluated on a single example, which could reflect the robustness against perturbing that particular example. Moreover, simulation results show our conclusion holds in higher rank cases (Figure 4). The challenge of generalizing the proof to higher Jacobian rank case is that we are no longer guaranteed that the leading eigenvectors of BPTT Jacobian coincide with the leading eigenvectors of an approximate rule Jacobian. Thus, it becomes much harder to relate $|\widehat{\lambda_1^J}|$ and $|\lambda_1^J|$. Rather than providing further proof, we provide an intuition for why our conclusion — where the convergence behavior between rules differs by their step length along the gradient direction — can hold in higher rank cases under Assumption 1.

**Assumption 1.** *Approximation error vector $\vec{e}$ (but not $\vec{g}$) lies orthogonal to the subspace spanned by the leading Hessian eigenvectors. Here, leading Hessian eigenvectors refer to the eigenvectors corresponding to the outlier Hessian eigenvalues in light of the well-known observation that there exists only a few large (outlier) eigenvalues and the rest are near zero [116, 117, 173]).*

The ramification of Assumption 1 is that $\vec{e}$ will lie in the subspace spanned by eigenvectors corresponding to tiny eigenvalues, making $H\vec{e}$ tiny. In the extreme scenario where $\vec{e}$ lies in the null space of $H$, $H\vec{e}$ would be 0. We verify this assumption numerically in Figure 10. We saw from the proof above that this assumption is automatically satisfied in the rank-1 Jacobian case. We remark that this assumption should not hold for stochastic gradient noise (SGN), as the SGN covariance matrix is well aligned with the Hessian matrix near a minima [122]. This could be why $\vec{e}$, unlike stochastic gradient noise, does not seem to be contributing much to escaping narrow minima.

We consider the case of small enough weight updates such that the loss surface can be approximated using second-order Taylor expansion. Thus, the loss change after one update becomes:

$$\Delta L \approx \Delta W^T \vec{g} + \frac{1}{2} \Delta W^T H \Delta W$$

$$= -\eta_B \vec{g}^T \vec{g} + \frac{1}{2} \eta_B^2 \vec{g}^T H \vec{g}, \quad \text{(for exact rule)} \tag{33}$$

$$\widehat{\Delta L} \approx \widehat{\Delta W}^T \vec{g} + \frac{1}{2} \widehat{\Delta W}^T H \widehat{\Delta W}$$

$$= -\eta_e \vec{\hat{g}}^T \vec{g} + \frac{1}{2} \eta_e^2 \vec{\hat{g}}^T H \vec{\hat{g}}. \quad \text{(for an approximate rule)} \tag{34}$$

We next focus on the first- and second-order Taylor terms ($T_1$ and $T_2$) for the exact rule as well as the terms ($\widehat{T_1}$ and $\widehat{T_2}$) for an approximate rule:

$$T_1 := \eta_B \vec{g}^T \vec{g}, \widehat{T_1} := \eta_e \vec{\hat{g}}^T \vec{g}, T_2 := \frac{1}{2} \eta_B^2 \vec{g}^T H \vec{g}, \widehat{T_2} := \frac{1}{2} \eta_B^2 \vec{\hat{g}}^T H \vec{\hat{g}},$$

and we note that first and second Taylor terms can determine how likely the update will be trapped in a local minimum:

$$
\begin{aligned}
\Delta L < 0 \text{ (enables descend)} &\quad \to T_1 > T_2 \\
\Delta L > 0 \text{ (restricts convergence)} &\quad \to T_1 < T_2 \\
\widehat{\Delta L} < 0 \text{ (enables descend)} &\quad \to \widehat{T_1} > \widehat{T_2} \\
\widehat{\Delta L} > 0 \text{ (restricts convergence)} &\quad \to \widehat{T_1} < \widehat{T_2}.
\end{aligned}
\tag{35}
$$

Given their central role in determining convergence, we compare these terms between exact gradient descent learning and an approximate rule. For the first Taylor term (T1), it is easy to see that:

$$
\vec{\tilde{g}}^T \vec{g} = \rho \vec{g}^T \vec{g}.
$$

For the second Taylor term (T2) and if $H$ is symmetric:

$$
\vec{\tilde{g}}^T H \vec{\tilde{g}} = \rho^2 \vec{g}^T H \vec{g} + 2\rho \vec{g}^T \underbrace{H \vec{e}}_{\approx 0} + \rho^2 \vec{e}^T \underbrace{H \vec{e}}_{\approx 0,\ \text{Assumption 1}}
$$

$$
\approx \rho^2 \vec{g}^T H \vec{g}.
\tag{36}
$$

To match the convergence behavior between exact gradient descent and an approximate rule (Eq. 35) on a (locally) second-order loss surface, we can make $(T_1, T_2)$ approximately equal to $(\widehat{T_1}, \widehat{T_2})$ by setting $\eta_B = \rho \eta_e$, which predicts our numerical results (Figure 4). We note that if Assumption 1 were not satisfied, then the above might not hold. We note in passing that if we can satisfy Assumption 1 without being near an optimum, then we may not need the negligible training error assumption.

# C   Additional Simulations

In the last paragraph in Discussion, we discussed how learning rate modulation could be one of the potential remedies used by the brain. We also explained how learning rate modulation could serve as a balance between the potential benefits of a large learning rate and the numerical stability issue mentioned shortly after the presentation of Figure 4 and Theorem 1 in Results. In Appendix Figure 5, we used a large learning rate early in training to prevent premature stabilization in sharp minima followed by gradual decay to mitigate the stability issue. With this remedy, we observed a reduction in the curvature of the converged solution and an improvement in generalization performance. This result also connects with the finding that sensory depletion during critical periods in training deep networks, which can be related to a small learning rate early in training, can impair learning and yield convergence to sharp minima [191]. However, it is important to note that this strategy does not correct the problem; the gap still exists compared to BPTT, suggesting room for further research.

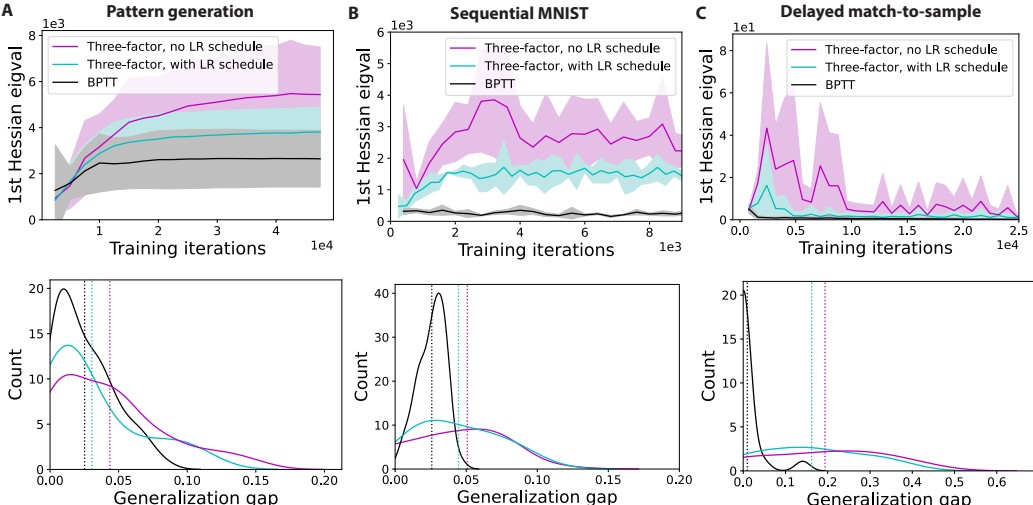

Figure 5: **Learning rate modulation as a possible remedy of the problem**. We increased the learning rate at the beginning of training to prevent the three-factor rule from stabilizing in sharp minima prematurely, followed by a gradual decay to prevent instability. The top panels show this strategy helps to reduce the curvature of the converged solution. The bottom panels show this leads to a slight improvement in the generalization gap (vertical lines denote distribution mean). However, it is important to note that this strategy does not correct the problem; the gap still exists compared to BPTT, suggesting room for further research. Plotting conventions follow that of the previous figures.

We present additional simulations referred to in the main text. In Figure 4, we attributed the convergence to high curvature regions to reduced along-gradient step length. In Appendix Figure 6, we confirm that such high curvature convergence indeed corresponds to worsened generalization performance, thereby linking reduced along-gradient step length to worsened generalization performance. As mentioned in the main text, due to the scale-dependence issue of Hessian spectrum [133], we also used scale-independent measures. For instance, we examined the power-law decay coefficient for the Hessian eigenvalues (Appendix Figure 7). We also looked at the recently proposed relative flatness measure [6] (Appendix Figure 8). These additional measures support the trends observed before: BPTT converges to lower curvature regions compared to the three-factor rule. We also observe that the tendency to approach high curvature regions seems to be a shared problem for temporal truncations of the gradient (Appendix Figure 9).

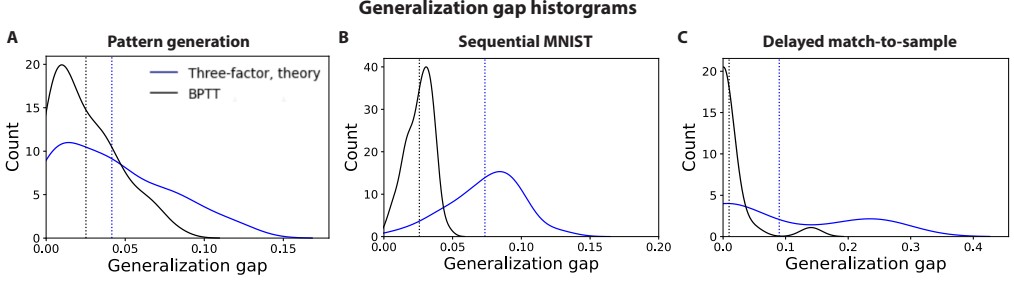

Figure 6: **Modified BPTT (three-factor, theory) resulted in worse and more variable generalization performance**. Here, we follow the convention of previous generalization gap histogram plots and investigated the generalization performance of modified BPTT (three-factor, theory) in Figure 4.

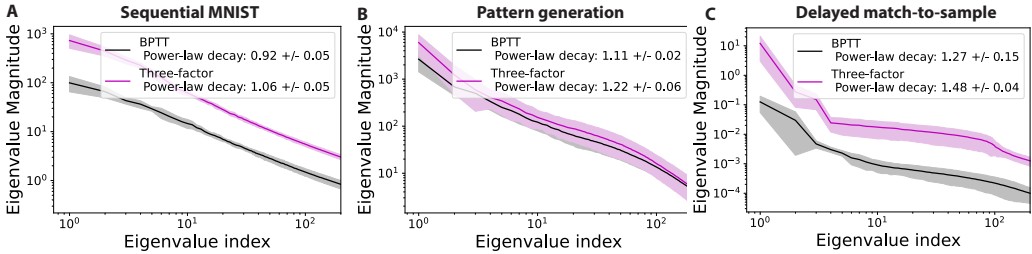

Figure 7: **Loss' Hessian eigenspectrum for the three-factor rule exhibits significantly steeper power-law decay compared to that of BPTT**. We fit a power-law function to the top 200 eigenvalues at the end of training and measure the decay parameter. Fitting to the top 50 or 100 eigenvalues resulted in similar trends. Solid lines/shaded regions: mean/standard deviation of eigenspectrum obtained at the end of training across five independent runs.

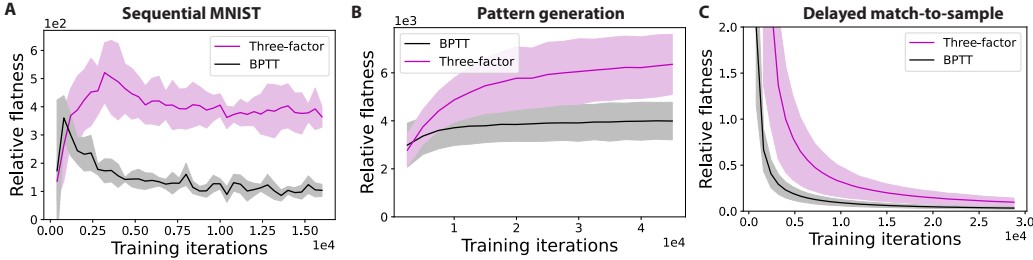

Figure 8: **Curvature preference behavior corroborated using relative flatness measure [6]**. Here, the trend is consistent with that of Figure 3. Note that the relative flatness measure can be computationally intensive for recurrent weights, so we computed it for readout weights. Plotting conventions follow that of previous figures.

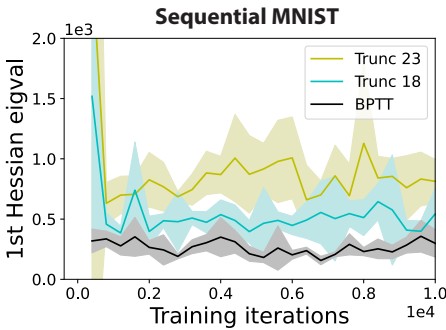

Figure 9: **Approaching high curvature regions seems to be a shared problem for temporal truncations of the gradient**. We repeat the analysis in Figure 3 for truncated BPTT (TBPTT). Here, "Trunc X" means X time steps are truncated during the gradient calculation. We observe that TBPTT tends to converge to high curvature regions. Plotting conventions follow that of previous figures.

In response to our discussion on the potential impact of noise direction (see explanation shortly after the presentation of Theorem 1, Discussion section and Appendix B.3), we confirm that the error vector $\vec{e}$ is significantly less aligned with the leading Hessian eigenvectors relative to the gradient vector $\vec{g}$ (Appendix Figure 10). As explained in Methods, we used SGD optimizer due to confounding factors in Adam optimizer that could convolute our matching step length analysis in Figure 4. We observe similar curvature convergence trends as in Figure 3 when we repeated the experiments with Adam optimizer in Appendix Figure 11.

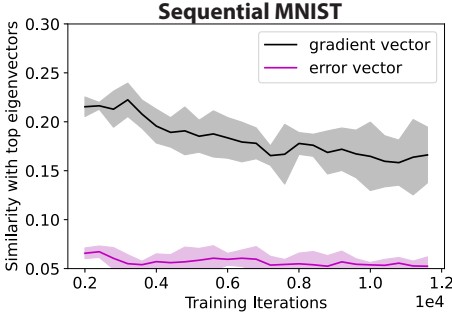

Figure 10: **Truncation error vector (compared to the gradient) is significantly less aligned with the top Hessian eigenvector subspace**. Following [172], we compute the cosine similarity between the error vector (of the three-factor rule) and a top Hessian eigenvector (averaged over the top 5 eigenvectors). The absolute value of the cosine similarity was taken. We observe weak alignment of the approximation error vector $\vec{e}$ with the leading Hessian eigenvectors. Similar trends were attained had we averaged over the leading 10 or 20 eigenvectors. Plotting conventions follow that of previous figures.

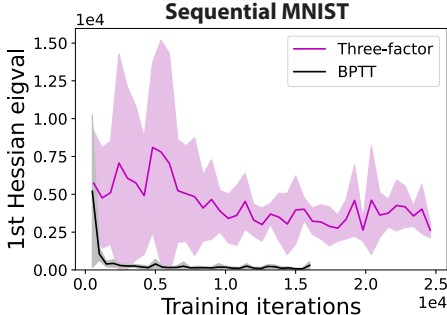

Figure 11: **Curvature convergence behavior also holds for Adam optimizer**. As explained in Methods, we used SGD optimizer due to confounding factors in Adam optimizer that could convolute our matching step length analysis in Figure 4. We observe similar trends as in Figure 3. Plotting conventions follow that of previous figures.

Finally, to further examine the correlation between leading Hessian eigenvalue and generalization performance (observed in Figure 2), we also observed such correlation correlation for runs with the learning rule fixed (Appendix Figure 12). For the matching step experiment in Figure 4, similar observations were also made when we repeated the experiment at three times the learning rate (Appendix Figure 12C). Moreover, we stopped BPTT early to match the test accuracy of the three-factor rule, and observed similar curvature convergence and generalization performance trends as previously (Appendix Table 1).

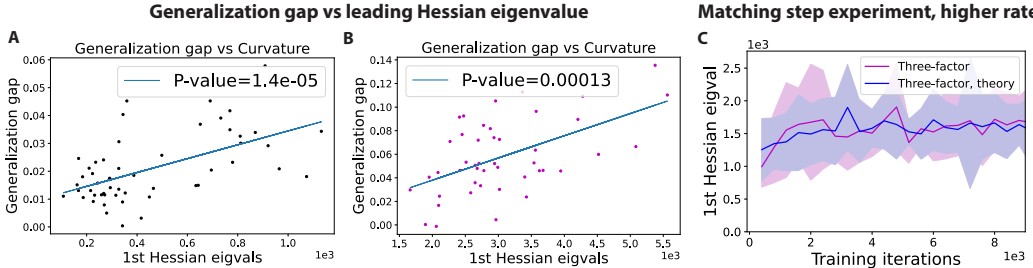

Figure 12: We repeat the generalization gap vs leading Hessian eigenvalue scatter plot in Figure 2 with the learning rule fixed for A) BPTT and B) the three-factor rule. As expected, a significant correlation between the generalization gap and leading Hessian eigenvalue is observed. Unlike Figure 2, where the hyperparameters were fixed for each rule (tuned using the procedure in Appendix A.3), the learning rate is varied here in order to get a wide enough curvature range to observe the correlation. C) The matching step experiments in Figure 4 were repeated here with the learning rate increased by three times for all rules, and the observation agrees with that in Figure 4. Plotting convention follows that of previous figures.

| Learning | Leading Hessian eigenvalue | Generalization gap |
|---|---|---|
| Three-factor | $2550 \pm 490$ | $0.5 \pm 0.3$ |
| BPTT, early stopping | $316 \pm 84$ | $0.2 \pm 0.1$ |

Table 1: BPTT stopped early to match the test accuracy of the three-factor rule for the sequential MNIST task. Higher generalization gap and leading Hessian eigenvalue is again observed for the three-factor rule, as expected. Each rule is repeated for five runs with different random weight initialization.