# OpenReview forum: "Beyond accuracy: generalization properties of bio-plausible temporal credit assignment rules"
_NeurIPS.cc/2022/Conference — NeurIPS 2022 Accept_

### Official Review · Reviewer_eejL · 2022-07-06

**Rating:** 8
**Confidence:** 4
**Soundness:** 4 excellent
**Presentation:** 4 excellent
**Contribution:** 4 excellent

**Summary:**

The paper investigates generalization and loss landscape curvature in RNNs trained with biologically-motivated learning rules. The paper finds empirically that generalization gaps are correlated with loss landscape curvature, with higher curvature correlating with larger generalization gaps. Next, the paper shows theoretically that approximations to gradient descent produce more curved solutions than gradient descent. This is because gradient descent approximations take smaller steps in the direction of the gradient and therefore are unable to escape highly curved local minima. The component of updates orthogonal to the gradient is less relevant to generalization. Experiments on multiple datasets support this explanation. Finally, the authors demonstrate empirically that appropriate learning rate scheduling in biologically-plausible learning rules can significantly enhance performance.

**Questions:**

This paper considers the generalization gap of different algorithms by roughly fixing their training performance and comparing their test performance. How would the results change if the test performance were fixed and the training performance were varied (by stopping the rules at different points during training, for example)?

Do generalization gaps correlate with Hessian eigenvalues in networks trained with the same rule?

How would a three-factor rule perform empirically when scaled to match the along-gradient update sizes of gradient descent?

At what point does numerical instability occur in biologically-plausible learning rules?

**Limitations:**

The authors adequately address the limitations of the work. As the authors note, they do not consider varying architectures and leave the interaction between architecture and generalization as future work to be investigated. Moreover, the authors note that the loss landscape curvature does not fully explain generalization.

**Strengths And Weaknesses:**

**Originality**
Although generalization in biologically-motivated learning rules has been studied previously, this paper provides surprising new insights into the nature of solutions found by biologically-plausible learning rules. Notably, to my knowledge, this paper is the first to quantitatively explain with high precision the relatively poor generalization performance of biologically-plausible learning rules (see Figure 4).

**Quality**
The theory is sound and well justified, and the experiments are generally comprehensive. The authors may want to consider some additional experiments to sharpen their claims.

First, the authors may want to investigate whether generalization gaps correlate with Hessian eigenvalues in networks *trained with the same rule* (rather than aggregating results across rules as done in Figure 2). This would help demonstrate the generality of the claim that generalization is tied to loss landscape curvature.

Second, the authors may want to consider an additional variant of the three-factor rule where the update sizes are scaled such that the gradient-parallel component matches that of gradient descent. As the authors argue in equation, this can cause numerical instability. Nevertheless demonstrating this empirically would be helpful and would nicely complement the "three-factor, theory" rule.

Finally, the paper seems to suggest that one of the main limitations of biologically-plausible learning rules is that using large update sizes for them leads to numerical instability while small update sizes leads to poorly generalizing local minima. Further explaining at which point this numerical instability occurs would be valuable (e.g. is the maximum stable learning rate limited by the eigenvalues of the Hessian?).


**Clarity**
The paper is well written. The figures are particularly well illustrated and clear. The mathematical notation is generally adequately introduced and used.

Some minor issues:
The W+ and W- notation in equations 7 and 8 is not explained.
Typo- menchmark on line 140.
Typo- assymptotic on line 1054.


**Significance**
The paper may be quite significant to the field of biologically-plausible learning in RNNs. One of the important takeaways from the paper is that learning rate modulation of biologically-plausible learning rules may be a key to good performance. This may spur further research into developing better learning rate schedules for biologically-plausible learning rules. Furthermore, this paper may have implications for deep learning theory more generally.

---

> ### Author Response · Authors · 2022-08-02
> **Response to eejL (2/2)**
>
> 5) **“At what point does numerical instability occur in biologically-plausible learning rules?”**
>
> This is an excellent question. Here are our main points in response to the reviewer’s question on which point this numerical instability occurs, and how to interpret this in biological contexts:
>
> - How can what we call numerical noise manifest in the brain? In digital computers, numerical instabilities are an issues because of rounding errors. While the analog nature of biology may prevent this, the same problems that lead to numerical instabilities, such as big ranges between quantities added or multiplied, remains an issue for biology since quantities must be stored in noisy activity patterns of neurotransmitter release. In other words, such noise could limit the "numerical precision" in the brain.
>
> - At which point such instability occurs could depend on homeostatic mechanisms in the brain, which could regulate quantities back to the "optimal operating range". It would certainly be an interesting future study to see if such homeostatic mechanisms can help alleviate the numerical stability issue in the brain, hence enabling larger "learning rates” to be used for bio-plausible temporal credit assignment rules so as to find flatter minima.
>
> As for which point this numerical instability occurs digitally:
>
> - It might be worthwhile to start by clarifying that there could be at least two kinds of instability: (A) the instability in the sense that the second order term in Taylor expansion of loss becomes significant relative to the first order term due to large step size; (B) numerical instability in the sense that numerical overflow can occur from large weight values in simulations.
>
> - For the explanation in our initial submission, we mainly considered (B) numerical instability. Large weight updates could result in accumulation of weight values. Of course, there are many regularization techniques to alleviate this issue (e.g. weight decay), but relative to BPTT, three-factor is more prone to this issue due to the additional orthogonal error component added to the weights, as explained in the manuscript.
>
> - If we are indeed considering instability in terms of (B), then determining at which point large weights could lead to numerical overflow would require careful calculation of how neuron activity propagates over time steps in relation to weight values. This calculation would be model and task dependent, and even precision dependent (e.g. 32-bit vs 64-bit). More importantly, this calculation would depend on the consistency of alignment of weight direction and update direction. If the two directions are consistently aligned, then the weight values just build up quicker.
>
> - If we are considering instability in terms of (A), then indeed as the reviewer said, the eigenvalues of the Hessian would matter, as it determines the magnitude of the second order loss term would indeed depend on the eigenvalues of the Hessian. In addition, how well aligned the update is with top Hessian eigenvectors also matters. As an extreme example, if the update lies in the null space of Hessian (i.e. aligned with eigenvectors with associated eigenvalues as 0), then the second order term would be 0. That ties nicely to a discussion point in the paper on how noise direction can affect generalization (please see also reviewer dP4V’s comments).
>
> - This might not be related to the reviewer’s question, but one thing to note is that the two kinds of instability can interact. For instance, achieving instability in (A) would require a large enough learning rate, so that the second order Taylor term of loss expansion can exceed the first order term. However, the instability in (B) can make it impossible to use large enough learning rates to achieve that desired instability in (A).
>
> To reflect these points above, we have now added a brief explanation in Results on numerical instability that occurs digitally, right after we first mentioned instability. After that explanation, we alerted the reader to discussions on numerical instabilities in the brain in Discussion, where we added a few sentences on numerical instabilities in the brain in the last paragraph of Discussion.

---

> ### Author Response · Authors · 2022-08-02
> **Response to eejL (1/2)**
>
> We would like to extend our gratitude to the reviewer for an excellent summary of our work, careful reading of our manuscript, and their specific suggestions on simulations to strengthen the paper. We would also like to thank the reviewer for appreciating several key contributions of the paper and shared vision on future directions. We are hopeful that the improvements and answers provided below will address the reviewer's concerns.
>
> 1) **“Some minor issues: The W+ and W- notation in equations 7 and 8 is not explained. Typo- menchmark on line 140. Typo- assymptotic on line 1054.”**
>
> We really appreciate the reviewer’s careful reading of our manuscript and finding these typos. We have fixed the typos and explained the W+ and W- notation.
>
> 2) **“Do generalization gaps correlate with Hessian eigenvalues in networks trained with the same rule?”**
>
> We thank the reviewer for the excellent suggestion to strengthen the argument. We have now created a scatter plot for leading Hessian eigenvalue and generalization gap with the same rule. **We repeated this for each of BPTT and three-factor rule**, and we added to plots to Appendix Figure 12 (and referred to in Results). We decided to perform the test on three-factor because the generalization curvature correlation has not been demonstrated previously for bio-plausible temporal credit assignment rules; we decided to perform the test on BPTT simply as a test of agreement with existing literature on generalization curvature correlation demonstrated for (S)GD. As expected, we found that generalization gap significantly correlate with leading Hessian eigenvalue in these experiments. Ideally we'd like to repeat this for all tasks but simulated one task due to time constraint.
>
> 3) **“This paper considers the generalization gap of different algorithms by roughly fixing their training performance and comparing their test performance. How would the results change if the test performance were fixed and the training performance were varied (by stopping the rules at different points during training, for example)?”**
>
> Again, we thank the reviewer for the great suggestion. We stopped BPTT when it reached the same test accuracy as three-factor and found that the leading Hessian eigenvalue and generalization gap is still significantly higher for three-factor. This should come as no surprise as we don’t see any point along the BPTT Hessian eigenvalue trajectory to match that of terminal Hessian eigenvalue attained by three-factor rule. We added these new results to Appendix Table 1 and referred to it in the Results section.
>
> 4) **“How would a three-factor rule perform empirically when scaled to match the along-gradient update sizes of gradient descent?”**
>
> This is a very interesting question. We should clarify that when we match the along-gradient update size, this may require us to increase the learning rate for three-factor rule by a factor of 20 times depending on the value of ρ (ρ vary depending on the task and model, but we’ve observed values ranging from 0.02 to 0.3). This significant increase in learning rate can quickly lead to numerical overflow, resulting in values of NaN in the network. This observation is typical when a very large learning rate is used. Once that happens, we cannot proceed with the training. We have added this explanation to the Results section in the main text.
>
> In response to the reviewer’s comment, we would still like to strengthen the matching step length experiment by doing additional runs but without numerical overflow, so we decided to go in the middle of the reviewer’s suggestion vs what was done in Figure 4: we repeated the experiment at three times the learning rate used in Figure 4. We again found that matching along-gradient update sizes lead to similar curvature convergence. This new plot can be found in Appendix Figure 12C and we referred to it in the Results section.
>
> **For response part (2/2), please proceed to our comment below.**

---

> ### Author Response · Authors · 2022-08-08
> **Request for discussion**
>
> Given that the author-reviewer discussion period is coming to a close soon, we kindly request the reviewer to let us know if our responses have resolved their concerns, and if there are any other questions that we can address. We are hopeful the reviewer will recognize that our recent work addresses initial concerns. We we are keen to further improve our paper in light of a constructive author-review discussion.

---

> > ### Comment · Reviewer_eejL · 2022-08-08
> > **Thank you for your reply**
> >
> > I appreciate the authors' careful response and additional experiments. I think the additional experiments help sharpen the claims made in the paper and increase its novelty. The explanations regarding numerical instability are particularly valuable. As the authors mention, additional regularizations can mitigate this numerical instability. Further numerical investigations into using various regularizations with biologically-plausible learning rules would be an interesting future direction, but may understandably be outside of the scope of this submission.
> >
> > Overall, I believe this paper can be a strong contribution to the conference. In light of the additional experiments and explanations, I have increased my rating for this submission.

---

> > > ### Author Response · Authors · 2022-08-09
> > > **Thank you**
> > >
> > > We are extremely grateful to this reviewer for careful reading of our response and taking that into consideration to revise their score. Moreover, we are very glad that the reviewer shares our excitement for future investigations into various regularizations with bio-plausible learning rules and how they could balance between potential benefits of large learning rates for generalization and numerical stability issues in the brain. We would like to extend our huge gratitude to the reviewer again for all their specific suggestions on simulations and discussion points that led to the improvement of this paper.

---

### Official Review · Reviewer_dP4V · 2022-07-08

**Rating:** 7
**Confidence:** 5
**Soundness:** 3 good
**Presentation:** 3 good
**Contribution:** 3 good

**Summary:**

The  paper aims to study the generalizability of biologically plausible learning rules for RNNs compared to backprop through time which is not bio-plausible. This work relies heavily on the relationship between the curvature of the loss landscape and the generalizability of the model: that wider minima are more noise tolerant and generalize better. Empirically it is shown that bio-plausible learning rules generalize worse than the common implausible ML learning rules. This is then theoretically explained as being due to the instability introduced by the truncation of the true backprop gradient being used by the bio-plausible rules. Specifically, to avoid diverging with the bio-plausible rules a smaller learning rate can be used which makes gradient descent more susceptible to converge to steeper minima.

**Questions:**

Assuming I have understood the work and not misrepresented the fact above, I have no questions for clarification at present. For the rebuttal period I would find it helpful for the authors to further address the concept of different kinds of noise impacting generalization differently.

**Limitations:**

The authors were clear that this study was not comparing bio-plausible learning rules but rather confirming that a well-established concept in ML carried to bio-plausible rules. This is the main limitation but it was clearly acknowledged. They also acknowledge that the relationship between curvature and generalization, while being the main point of study in this work, is messy and state that other factors may be at work for generalization.

**Strengths And Weaknesses:**

# Strengths
## Originality
This paper addresses a topic which I have not seen in the computational neuroscience literature and does so with care for both the machine learning and computational neuroscience considerations. I think the primary novelty of this work however comes in at the end with the finding that the noise from truncation does not affect generalization in the same manner as noise from smaller batch sizes for example (truncation noise hinders generalization while other forms of noise improve generalization by driving learning out of sharp minima). This is a new consideration to me and lead me to start thinking around whether the consistency of noisy direction of learning plays a role.

## Quality
There are no points during the paper which stand out as being unreasonable, and from a scientific method point of view the work appears sound, with each step being justified either from a comp-neuro or ML perspective. Balancing the two topic was done well and I think that is a potentially understated positive for this work.

## Clarity
The paper is well written and I found it to be clear and concise. The figures are well made and helpful.

## Significance
The work appears significant from two perspectives. Firstly, work at the intersection of comp-neuro and ML is important and potentially very fruitful, however, not very common (at least not at the level of insight for both fields offered by this paper). Thus, I think this is significant to the comp-neuro community as a step towards introducing some of the more theoretical ML concepts such as loss landscape curvature. Secondly, this appears significant as it brings up the interesting concept that some noise may help generalization while other forms of noise hurt generalization. It does so in a manner which is also very useful from a learning stability perspective which also uses the Hessian matrix [1]. Thus from a theoretical ML perspective I think this work offers interesting insight and potential inspiration for future work.

# Weaknesses
## Originality and Significance
It must be noted that this work relies heavily of previous work in theoretical ML and tends to confirm what is already known (wider minima generalize better). That said, I think just because the results are not surprising does not mean they are not valuable. I am glad that this work has been done and acknowledge that it was worthy of confirming that the findings from ML carried towards bio-plausible learning rules. Indeed it seems to me that this work can still inspire future directs of ML work as I point out above. Thus I feel the strengths for originality and significance do outweigh this weakness.

That said, if the authors were to add to the discussion at the end about truncation noise reducing the stable step size I would be inclined to increase my rating to a 7 or an 8. The one citation below may be of use for this purpose. I do think that this paper is worth of acceptance and is of general interest to the machine learning and comp-neuro communities.

[1] Cohen, Jeremy M., et al. "Gradient descent on neural networks typically occurs at the edge of stability." arXiv preprint arXiv:2103.00065 (2021).

---

> ### Author Response · Authors · 2022-08-02
> **Response to dP4V (2/2)**
>
> 2) **“For the rebuttal period I would find it helpful for the authors to further address the concept of different kinds of noise impacting generalization differently.”**
>
> Again, we would like to thank the reviewer for the stimulating discussion! Here are our comments in addition to our response to the previous comment above. To begin discussing the impact of different kinds of noise, we can start by deciding on the aspects of noise we would like to focus on. One aspect could be **direction**, which has been discussed in the manuscript and briefly mentioned in the previous comment.
>
> However, another aspect we did not discuss in the manuscript but the reviewer brought up could be **consistency**, which could also play an important role in generalization. Intuitively, if noise is constantly biased toward one direction, then the weight could build up quickly along that direction, and the increased weight norm can significantly limit the learning rate to avoid numerical instability; and we have discussed how small learning rate can hurt generalization in the manuscript. However, if the noise direction fluctuates and cancels out each other across update steps, then weight may not build up as much so the numerical instability issue is less of a concern.
>
> Echoing the reviewer’s comments, we look forward to seeing future studies on more characterization of different sources of noise that appear in biological systems (e.g. learning rule ese directions are). These noises can have different structures than the kinds of noise in optimization. We believe the ML community provides valuable tools to examine how different noises from neural systems can impact generalization, and if there are certain biological noise that make them more favorable for generalization. We have now added discussion points to the Discussion section reflecting this point.
>
> **References:**
>
> [1] Cohen, Jeremy M., et al. "Gradient descent on neural networks typically occurs at the edge of stability." arXiv preprint arXiv:2103.00065 (2021).
>
> [2] Aitor Lewkowycz, Yasaman Bahri, Ethan Dyer, Jascha Sohl-Dickstein, and Guy Gur-Ari. The large learning rate phase of deep learning: the catapult mechanism. arXiv preprint arXiv:2003.02218 (2020).
>
> [3] Stanisław Jastrzębski, Maciej Szymczak, Stanislav Fort, Devansh Arpit, Jacek Tabor, Kyunghyun Cho*, and Krzysztof Geras*. The break-even point on optimization trajectories of deep neural networks. In International Conference on Learning Representations (2020).
>
> [4] Justin Gilmer, Behrooz Ghorbani, Ankush Garg, Sneha Kudugunta, Behnam Neyshabur, David Cardoze, George Edward Dahl, Zachary Nado, and Orhan Firat. A loss curvature perspective on training instabilities of deep learning models. In International Conference on Learning Representations (2022).
>
> [5] Sanjeev Arora, Zhiyuan Li, and Abhishek Panigrahi. Understanding gradient descent on edge of stability in deep learning. arXiv preprint arXiv:2205.09745 (2022).truncation, data noise) and their bias/variance properties (noise direction and how consistent th

---

> > ### Comment · Reviewer_dP4V · 2022-08-05
> > **Rebuttal Response**
> >
> > Thank you to the authors for their response. I have checked the added content of the paper and it appears correct (I appreciate the authors making the changes clear and this process easy). I think the authors have understood the point of my suggestion to discuss learning rate in terms of stability; and the added paragraphs in the Discussion are helpful and sufficient for this work. I would be very interested to see a more theoretical approach to understanding the relationship, however to me that is clearly beyond the scope of this work. The point on there being two types of instability is quite nuanced and I appreciate the authors being clear on that in the rebuttal.
> >
> > My current score (of 6) still reflects my sentiments on the work and so I will leave it as such for the moment (I will follow the discussion with my fellow reviews and raise any points if necessary).

---

> > > ### Author Response · Authors · 2022-08-05
> > > **thank you**
> > >
> > > We thank the Reviewer for their response and follow-ups. We agree that there is further exciting theoretical work ahead to address stability, and that this falls outside the scope of the current paper, which offers considerable contributions as it stands.
> > >
> > > As the reviewer points out, we too believe the paper would be a great contribution to the scientific community as it presents innovative work at the intersection of ML theory and computational neuroscience. Furthermore, we think that NeurIPS22 is an ideal venue and time to do so. We feel the reviewer accurately recognizes the value and originality of such cross-disciplinary work, which can often suffer from shortcomings in review processes as it spans distant areas of expertise. As such, we would be extremely grateful if the reviewer would identify further areas we could improve within the paper's present scope to facilitate a score increase. We understand from comments the reviewer is supportive of this work being published and would be grateful for the opportunity to earn their continued support in the final stages of reviews.
> > >
> > > very best regards,

---

> > > > ### Comment · Reviewer_dP4V · 2022-08-08
> > > > **Further Improvements/Impacts**
> > > >
> > > > As requested I have given some more thought to improvements that can be made in the current scope of the work. There is not much in the current scope to be improved (I include reducing the amount of dependence on prior work a change of scope). You can see my original review for my feelings on the originality/quality/significance as they have not changed.
> > > >
> > > > What has changed upon reflection is that I may have under-appreciated the quality of the Appendix. While not usually something which should impact a review too much, in the context of this work I think it is relevant to consider. Specifically, the reliance on prior work, while hindering the main paper slightly, is a real benefit in the Appendix as you have unified many fields (ML,Neuro,Geometry) well. I could see the Appendix introducing many readers to one of the fields which is not the one that drew them to the paper in the first place. For example, the equality of the Fisher Information Matrix (outer product of the Jacobian) and Hessian at a minimum in the loss landscape (a key concept in Information Geometry) is present in Appendix B, while Appendix A does a good job of introducing the reader to truncated learning rules and why they are biologically plausible. That said, the Appendix as it stands could be improved.
> > > >
> > > > Here are my suggestion:
> > > > 1. Please check the spelling and grammar in the Appendix as there are a few mistakes.
> > > > 2. Flesh out Appendix C. Currently it is just a collection of Figures with captions. Please write this as you would a usual section in a main body of work. It does not have to be much, but a bit more of a formal section than it currently is.
> > > > 3. You mention in the last paragraph of the Introduction (revised edition) that "We also discuss potential remedies used by the brain (Appendix Figure 5)". Unless something is getting lost in the edits I don't think Appendix Figure 5 covers this, at least not enough to be mentioned like it is in the main paper. That point is interesting and I would like to see those remedies, I assume learning rate modulation is the main point of this, but how learning rate modulation may be done in the brain then should be mentioned (ideas around plasticity may be new to some ML readers and I think your clarity on these concepts in the Appendix will serve them greatly).
> > > > 4. Reference points in the Appendix more clearly in the main paper. In general I think this is done well, but I think in this case your clarity in helping a reader navigate to the information they find interesting but unfamiliar in the Appendix will help.
> > > >
> > > > I trust the authors will make the necessary changes and I do not intend any of these to be too drastic. But rather the authors can make small changes in line with these suggestions and I do believe it will go a long way. Please let me know when a new version of the paper is available and I will increase my score.

---

> > > > > ### Author Response · Authors · 2022-08-09
> > > > > **Response to Further Improvements/Impacts (2/2)**
> > > > >
> > > > > 3. **You mention in the last paragraph of the Introduction (revised edition) that "We also discuss potential remedies used by the brain (Appendix Figure 5)". Unless something is getting lost in the edits I don't think Appendix Figure 5 covers this, at least not enough to be mentioned like it is in the main paper. That point is interesting and I would like to see those remedies, I assume learning rate modulation is the main point of this, but how learning rate modulation may be done in the brain then should be mentioned (ideas around plasticity may be new to some ML readers and I think your clarity on these concepts in the Appendix will serve them greatly).**
> > > > >
> > > > > We agree with the reviewer that it is unclear from Appendix Figure 5 how such remedies can be implemented. **On top of Appendix Figure 5, we should have also referenced the last paragraph of Discussion section**, where we discussed experimental predictions and how learning rate modulation could be implemented in the brain, *“We conjecture that neuromodulatory mechanisms could be coupled with these learning rules to improve the convergence behavior through our scheduled learning rate experiments (Appendix Figure 5), where an initial high learning rate could prevent the learning trajectory from settling in sharp minima prematurely followed by gradual decay to avoid instabilities. One possible way to realize such learning rate modulation could be through serotonin neurons via uncertainty tracking, where the learning rate is high when the reward prediction error is high (this can happen at the beginning of learning) [169]. Since the authors of [169] showed that inhibiting serotonin led to failure in learning rate modulation, we conjecture that such inhibition might have an impact on the generalization performance of learning outcomes.”*
> > > > >
> > > > > Therefore, **we have replaced the sentence in Introduction that the reviewer quoted with** *“In the last paragraph of the Discussion section, we discuss potential remedies implemented by the brain and provide preliminary results (Appendix Figure 5)”* in order to direct the reader to the Discussion paragraph also.
> > > > >
> > > > > Additionally, In response to this and the previous comment, we also summarized the key points above in the text that surrounds Figure 5 in Appendix.
> > > > >
> > > > > 4. **Reference points in the Appendix more clearly in the main paper. In general I think this is done well, but I think in this case your clarity in helping a reader navigate to the information they find interesting but unfamiliar in the Appendix will help.**
> > > > >
> > > > > For this comment, we have focused on making Appendix B (Geometry and ML theory) and Appendix A (bio-plausible temporal credit assignment rules) more visible in the main text, so as to facilitate the introduction of readers to these different areas.
> > > > >
> > > > > For Appendix B, in response to the reviewer’s remark *“For example, the equality of the Fisher Information Matrix (outer product of the Jacobian) and Hessian at a minimum in the loss landscape (a key concept in Information Geometry) is present in Appendix B”*, we have referred to that in Discussion, specifically as the ending sentence of the discussion paragraph on step size stability that the reviewer requested in an earlier comment, *“This increased stability is closely tied to Theorem 1 --- which predicts a greater dynamical stability for (the weight update difference equation of) three-factor rules --- due to the correspondence between loss' Hessian matrix and the Jacobian matrix of the weight update difference equation (the correspondence is explained in Theorem 1 proof in Appendix B).”* On top of that, we have alerted the readers in Introduction to check out Appendix B, *“we encourage the reader to visit Appendix B for the Theorem Proof and discussion on loss landscape geometry”*.
> > > > >
> > > > > For Appendix A, we have alerted the readers in Introduction to visit Appendix A.2 to learn more about bio-plausible temporal credit assignment rules, *“for in-depth explanation of these bio-plausible temporal credit assignment rules, please visit Appendix A.2 for how these rules are implemented and why they are bio-plausible”*.

---

> > > > > ### Author Response · Authors · 2022-08-09
> > > > > **Response to Further Improvements/Impacts (1/2)**
> > > > >
> > > > > We would like to extend our huge gratitude to the reviewer for not only appreciating the key points and contributions in our Appendix, but also providing specific and feasible suggestions that can make a big impact on the accessibility of this paper. As the reviewer correctly pointed out, the Appendix provides important supporting information that helps to introduce readers to unfamiliar areas, so it is crucial to improve the visibility and presentation of these sections. Thus, we have taken this reviewer’s suggestions very seriously and outlined our updates below. We are hopeful that after having implemented this reviewer’s suggestions, the accessibility and impact of our manuscript are going to be significantly improved.
> > > > >
> > > > > 1. **Please check the spelling and grammar in the Appendix as there are a few mistakes**.
> > > > >
> > > > > We are grateful for the suggestion. Indeed, we have indeed caught multiple grammar mistakes and typos in Appendix. Our updates include but are not limited to the following. A further thorough pass for typos will be done before the camera-ready version if this paper were accepted.
> > > > >
> > > > > - Appendix A.2: changed "Thus, the factor $\frac{\partial h_{l,t}}{\partial W_{h, ij}}$ and it poses" to "Thus, the factor $\frac{\partial h_{l,t}}{\partial W_{h, ij}}$ poses"
> > > > > - Appendix A.2: changed "expensive nonlocal term" to "expensive nonlocal terms" (should be plural)
> > > > > - Appendix A.3: changed "we note that comparisons BPTT and approximate rules are done at" to "we note that comparisons between BPTT and approximate rules were done at"
> > > > > - Throughout Appendix A: changed to past tense at multiple places when we were describing Methods
> > > > > - Appendix B.2: changed "how close the magnitude of the leading Jacobian eigenvalues are to 1" to "how close the magnitude of the leading Jacobian eigenvalue is to 1" (should be singular)
> > > > > - Appendix B.2: changed “In the extreme scenario where the loss is convex, there is just one minimum, so the question of minima preference becomes irrelevant” to "If the loss were convex, there would just be one minimum and the question of minima preference would become irrelevant" (fixed verb tense)
> > > > > - Appendix B.3: changed "higher rank case" to "higher rank cases" at multiple places (should be plural)
> > > > > - Appendix B.3 (also B.1): added articles, e.g. "an approximate rule" instead of just "approximate rule", at multiple places
> > > > > - Appendix B.3: changed "for going further down" to "enables descend" to be more concise
> > > > > - Appendix B.3: changed "for a (locally) second order loss surface" to "on a (locally) second order loss surface)"
> > > > > - Appendix C: Figure 6 caption, fixed an unclosed parenthesis
> > > > > - Appendix C: Figure 7 caption, changed “Similar results were obtained if we fit to the top 50 or 100 eigenvalues” to "Fitting to the top 50 or 100 eigenvalues resulted in similar trends" (fixed verb tense)
> > > > > - Appendix C: Figure 9 caption, changed "we observe TBPTT tend to" to "we observe that TBPTT tends to"
> > > > > - Appendix C: Figure 11 caption, changed "use" to "used" (fixed verb tense)
> > > > >
> > > > > 2. **Flesh out Appendix C. Currently it is just a collection of Figures with captions. Please write this as you would a usual section in a main body of work. It does not have to be much, but a bit more of a formal section than it currently is**.
> > > > >
> > > > > We absolutely agree with the reviewer that having surrounding texts in Appendix C would improve the formality. We have now added texts throughout Appendix C and made sure that we referred to every single figure in Appendix C in the same manner as we would in a main body of work. Again, we colored these texts in blue.

---

> > > > > ### Author Response · Authors · 2022-08-09
> > > > > **discussion period coming to an end...**
> > > > >
> > > > > We would like to take this final moment to thank the reviewer once more for a stimulating exchange and encouraging feedback. We hope the reviewer will be satisfied with the changes made to the paper and appendix, as they suggested. We would be grateful if this would enable a score increase.
> > > > >
> > > > > best regards,
> > > > > the authors

---

> ### Author Response · Authors · 2022-08-02
> **Response to dP4V (1/2)**
>
> We would like to extend our huge gratitude to the reviewer for an excellent summary of our work, careful reading of our manuscript, and most importantly, for their astuteness in highlighting one of the most important points of the paper: noise from truncation does not affect generalization in the same manner as stochastic gradient noise. We also thank the reviewer for the shared vision and the comments on interesting future directions that this work can inspire. We are hopeful that the improvements and answers provided below will address the reviewer's concerns.
>
> 1) **Add to the discussion at the end about truncation noise affecting the stable step size; the one citation below [1] may be of use for this purpose**
>
> We are grateful that the reviewer is creating this very interesting discussion thread. We have also reflected points below in Discussion in response to the reviewer's great comment. We also would very much appreciate it if the reviewer could please be so kind as to let us know if we misunderstood anything from the reviewer's comment.
>
> If we understood correctly, the reviewer would like us to discuss how truncation noise affects the threshold at which the learning rate (or step size) would be flipped from being stable to unstable. If our understanding is correct, a step size is defined to be stable (on page 5 in [1]) if it is small enough such that the sharpness never rises to 2/$\eta$. For a full batch gradient descent case (as in [1]), sharpness crossing 2/$\eta$ would correspond to when the 2nd order Taylor expansion term of loss catches up to the 1st and learning could "catapult" into a flatter region to accommodate the step size (using the language of [1-2]). Our results suggest that truncation noise can make it harder for the “catapult'' behavior to happen. If the noise is not aligned with the few leading Hessian eigenvectors with large outlier eigenvalues but aligned with the eigendirections with negligible eigenvalues, then it can only have limited contribution to the 2nd Taylor term (see also Appendix Eq. 33-36); this ties nicely back to the reviewer’s and our remark on how noise direction matters. Moreover, the noise term demands a smaller learning rate to be used to avoid numerical issues (as explained in the paper). Because the noise is not adding much to the 2nd Taylor term and demands a smaller learning rate, it would weaken the 2nd Taylor term and hence make learning harder to “catapult” into a flatter region, thereby increasing the threshold for step size stability. This is all consistent with the results on convergence to high-curvature loss landscape regions for bio-plausible temporal credit assignment rules seen in this study.
>
> We would also like to thank the reviewer for bringing up [1], as the series of “catapult” behavior mentioned in [1] and [2] — happen in this so-called Edge of Stability (EoS) regime — seems closely related to our results. Along with [1], we have cited a few other works [2-5] in the updated manuscript that touched upon EoS explicitly or implicitly.
>
> Also, we are not sure if this is relevant to the reviewer's comment, but to potentially clarify our above paragraphs better, we would like to mention instability could mean at least two things: (A) the instability in the sense that the second order term in Taylor expansion of loss becomes significant relative to the first order term due to large step size; (B) numerical instability in the sense that numerical overflow can occur from large weight values in simulations. For our explanation (in the initial submission) on how orthogonal noise can contribute instability, we mainly considered (B) numerical instability: large weight updates could result in accumulation of weight values. Of course, there are many regularization techniques to alleviate this issue (e.g. weight decay), but relative to BPTT, three-factor is more prone to this issue due to the additional orthogonal error component added to the weights, as explained in the manuscript. One thing to note is that the two kinds of instability can interact. For instance, achieving the instability in (A) would require a large enough learning rate, so that the second order Taylor term of loss expansion can exceed the first order term. However, the instability in (B) can make it impossible to use large enough learning rates to achieve that desired instability in (A).
>
> **For response part (2/2), please proceed to our comment below.**

---

### Official Review · Reviewer_qy4u · 2022-07-14

**Rating:** 6
**Confidence:** 3
**Soundness:** 2 fair
**Presentation:** 2 fair
**Contribution:** 3 good

**Summary:**

The authors investigate the difference in the generalization gap in RNNs trained to minimize MSE loss with backpropagation through time and truncated BPTT. The authors claim the difference is caused by the landscape curvature of the loss, specifically, truncate BPTT approaches high-curvature regions in the synaptic weight space. The authors propose a theoretical argument to explain this phenomenon based on the first Hessian eigenvalue. The authors claim this result holds for all existing bio-inspired learning rules in RNNs and for different loss functions.

**Questions:**

Were potential remedies tested?

In figure 2 (a, b, c), does random initialization refer to weight initialization? Are all parameters of the learning algorithm held constant or are the distributions computed across all parameter configurations, e.g., learning rate ..etc? It appears that truncated BPTT can achieve the same generalization gap as BPTT. In the case of sMNIST it appears that it achieves zero generalization gap with a greater probability than full BPTT.

**Strengths And Weaknesses:**

Strengths:

* The authors address the important issue of how biological systems learn and adapt to stimuli. Specifically, the authors address that RNNs trained with truncated BPTT have a greater generalization gap than the full BPTT on three existing tasks.
* The authors focus on a well-defined metric, the 1st Hessian eigenvalue, in order to quantify their results.
* The authors provide a theoretical analysis that suggests a cause for the difference in the generalization gap between full BPTT and truncated BPTT

Weaknesses:
* The authors generalize their claims to all biologically-inspired learning rules but limit their study to three specific learning rules and three tasks. Specifically, the authors focus on truncated BPTT in RNNs on three tasks with MSE loss. These results may hold only for truncated backpropagation through time but not for all biologically-plausible learning rules. The style implies these results hold for all biologically-inspired learning rules and all loss functions. If this is true, it should be presented in the main text.
* The authors claim their results generalize beyond MSE loss but do not demonstrate this empirically.
* The authors make claims of potential remedies, such as using larger learning rates early in training but do not justify this claim theoretically or demonstrate it empirically.
* It is uncertain and not addressed if the generalization gap can be reduced by fine-tuning existing truncated BPTT algorithms.
* A description of training parameters and procedure used for the biologically-plausible learning rules is not presented in the main test.
* The authors cite almost 100 articles in section 2.1. Though these articles may be relevant to biologically-plausible learning in general it is not certain they are directly relevant to networks trained with truncated BPTT which is the focus of this study.

---

> ### Author Response · Authors · 2022-08-02
> **Response to qy4u (2/2)**
>
> 5) **“The authors cite almost 100 articles in section 2.1. Though these articles may be relevant to biologically-plausible learning in general it is not certain they are directly relevant to networks trained with truncated BPTT which is the focus of this study.”**
>
> We thank the reviewer for this comment.
>
> There is a lot of relevant literature on generalization gap, loss landscape, bio-plausible rule and RNN model for the brain. **To our best knowledge, our work is one of the first that introduces theoretical ML concepts such as loss landscape curvature (and how that affects generalization) to the computational neuroscience community** (reviewer dP4V also acknowledges this). As such, we tried to be as comprehensive in our citation as we can to cover the different research areas that inspired this work. We apologize that it turned out to be crowded, but we felt that it’s a good reference service to the community.
>
> We agree with the reviewer that existing bio-plausible learning rules for RNN, and how they are all truncation-based (to our knowledge), should be made more prominent in Related Works. To reflect that, we have added the following to Section 2.1, *“These existing bio-plausible rules for training RNNs [11-13] are truncation-based (which is the focus of this study), so that the untruncated terms of the gradient can be assigned with putative identities to known biological learning ingredients:...”*
>
> 6) **“The authors generalize their claims to all biologically-inspired learning rules but limit their study to three specific learning rules and three tasks. Specifically, the authors focus on truncated BPTT in RNNs on three tasks with MSE loss. These results may hold only for truncated backpropagation through time but not for all biologically-plausible learning rules. The style implies these results hold for all biologically-inspired learning rules and all loss functions. If this is true, it should be presented in the main text.”**
>
> We thank the reviewer for the great suggestion. **Indeed, we focus on temporal credit assignment rules, and we specified that in the title in our initial submission**. To our knowledge most if not all existing bio-plausible temporal credit assignment rules are truncation-based, so that the approximate gradient would involve only terms that can be assigned with putative identities to known biological learning ingredients. In response to the reviewer’s suggestion, we have now further clarified this point throughout the manuscript.
>
> In addition, we have added a discussion sentence for future investigations to our analysis to other bio-plausible learning systems beyond RNNs, *“This indicates that noise with different properties (e.g. direction) could affect generalization differently, thereby motivating future investigations into applying our curvature-based analysis to broader range of bio-inspired learning systems and examine biological noise with different properties”*. On that note, since our work is one of the first that introduces theoretical ML concepts such as loss landscape curvature (and how that affects generalization) to the computational neuroscience community (to our knowledge and also mentioned by reviewer dP4V), we hope our work will inspire more future studies that leverage the remarkable progress from the ML community to study learning and generalization for a broad range of neural mechanisms and systems.
>
> Regarding loss function, we used cross-entropy loss for classification tasks. Please see one of our earlier responses. It’s true that we only ran three tasks empirically, but we expect our conclusions to generalize beyond these three tasks due to our theory.
>
> 7) **“In figure 2 (a, b, c), does random initialization refer to weight initialization? Are all parameters of the learning algorithm held constant or are the distributions computed across all parameter configurations, e.g., learning rate ..etc? It appears that truncated BPTT can achieve the same generalization gap as BPTT. In the case of sMNIST it appears that it achieves zero generalization gap with a greater probability than full BPTT.”**
>
> We thank the reviewer for the astute observation. Yes, indeed it is possible for these truncation-based bio-plausible rules to achieve the same generalization gap as BPTT for some runs. However, the focus here is not to look at a few runs, *but the overall trend across many runs*. We hope the scatter plot (2d-f) can clarify the overall trend (worse and more variable generalization performance) and we are also going to make a note of this point in the caption.
>
> For the same learning rule, the same set of hyperparameters are used across runs. This is because hyperparameters should be tuned during the validation step before model deployment (please refer to Appendix A for parameter tuning details).
>
> Yes, “random initialization” refers to weight initialization. Thanks to the reviewer’s note, we have now clarified that in the caption.

---

> ### Author Response · Authors · 2022-08-02
> **Response to qy4u (1/2)**
>
> We would like to extend our gratitude to the reviewer for their valuable suggestions on how to improve the clarity of the paper. **We are hopeful that the presentation of our manuscript has been significantly improved after incorporating reviewer qy4u’s comments.**
>
> We would also like to make a general remark that due to the nine-page limit, we unfortunately had to include a lot of content (e.g. hyperparameter tuning, Theorem proofs, additional simulations) in the Appendix. In response to the reviewer’s comments, we incorporated direct references to Appendix content in the main text, in order to make such content more visible. Below, we directly address the reviewer's points, and describe related changes made to the manuscript.
>
> 1) **“The authors claim their results generalize beyond MSE loss but do not demonstrate this empirically.”**
>
> We used cross-entropy loss for the sequential MNIST task and binary cross-entropy loss for the delayed match-to-sample task. However, we understand where the confusion comes from. In the main text, we provided only the equation for MSE loss, although we provided provided the equation for cross-entropy loss in Appendix and alerted to reader to find that in the main text (please see *“We consider cross-entropy loss for classification tasks (Methods in Appendix A)”* in Section 3.1). In response to the reviewer’s point, we have added an equation for classification loss the the main text right beneath the MSE loss equation, in order to avoid this confusion.
>
> 2) **“The authors make claims of potential remedies, such as using larger learning rates early in training but do not justify this claim theoretically or demonstrate it empirically… Were potential remedies tested?”**
>
> Again, we apologize if this was not clear. We have tested potential remedies, which can be found in Appendix Figure 5. We wished to include the figure in the main text, but unfortunately we had to move it due to the page limit. To make Appendix Figure 5 more prominent, we have also referred to it in Introduction. In our initial submission, this figure and the results of empirical testing was referenced in Results and Discussion.
>
> 3) **“A description of training parameters and procedure used for the biologically-plausible learning rules is not presented in the main test.”**
>
> Indeed, detailed training procedures (including hyperparameter tuning) are not in the main text. We moved training details to Appendix A due to the nine-page limit. We have alerted the reader in Results to look for training details in Appendix A.
>
> 4) **“It is uncertain and not addressed if the generalization gap can be reduced by fine-tuning existing truncated BPTT algorithms.”**
>
> All of our runs have been hyperparameter tuned and training details can be found in Appendix A. We apologize if this detail is not clear, and this is now clearly stated in the revised main text.
>
> In addition, we explained the cause for our empirical observations using Theorem 1 and Figure 4, which suggest that the observed differences between rules cannot be reduced by fine-tuning learning rates. Please see our explanation on Figure 4 and Theorem 1 in Results. We have now added the following sentence to that section to clarify this point: *“Because of the numerical issues associated with increasing the learning rate for such rules, the differences in generalization and curvature convergence between learning rules cannot be reduced by fine-tuning the learning rate”*.
>
> **For response part (2/2), please proceed to our comment below.**

---

> ### Author Response · Authors · 2022-08-08
> **Request for discussion**
>
> Given that the author-reviewer discussion period is coming to a close soon, we kindly request the reviewer to let us know if our responses have resolved their concerns, and if there are any other questions that we can address. We are hopeful the reviewer will recognize that our recent work addresses initial concerns. We we are keen to further improve our paper in light of a constructive author-review discussion.

---

### Official Review · Reviewer_22dn · 2022-07-26

**Rating:** 5
**Confidence:** 3
**Soundness:** 4 excellent
**Presentation:** 3 good
**Contribution:** 2 fair

**Summary:**

- This paper investigates the ability of existing bio-plausible temporal credit assignment rules to generalize. Finding that training with bio-plausible learning rules results in models that have larger and more variable train/test generalization gaps.
- It then compares the loses’ hessian eigenspectrum of bio-plausible and SoTA learning rules finding that bio-plausible rules tend to approach high curvature regions in synaptic weight space
- It suggest an explanation for bio-plausible rules preference for high curvature regions based on worse alignment to the true gradient

**Questions:**

Are there other advantages to bio-plausible methods or advantages in terms of generalization can only be seen on more complex tasks?

**Limitations:**

yes

**Strengths And Weaknesses:**

 Strengths
- This paper is well written and clear
- It analyzed several learning rules and the results support the conclusion that bio-plausible methods don't generalize well
- The theoretical results offer interesting insights about the relationship between curvature and gradient alignment

Weakness
- It is well known that bio-plausible methods generally perform worse than SOTA back propagation. This work provides further empirical evidence that this is the case, but I do not find these results very surprising.

---

> ### Author Response · Authors · 2022-08-02
> **Response to 22dn (2/2)**
>
> 2) **“Are there other advantages to bio-plausible methods or advantages in terms of generalization can only be seen on more complex tasks?”**
>
> We thank the reviewer for the excellent question. We know the brain excels at generalization compared to most SoTA artificial systems on several tasks, so it’s logical to ask if there are advantages to bio-plausible methods not seen in this study, perhaps on more complex tasks.
>
> Before answering the question directly, we would like to remind the reviewer that there are a number of biological ingredients not explored in our study, notably architecture. As framed in introduction and explained in discussion, this paper examines learning rules in isolation as a starting study for bringing theoretical ML concepts such as loss landscape curvature to study biological learning systems. We hope our work can inspire many exciting future studies that apply our analysis to examine the impact of a diverse array of neural circuit elements on generalization and understand why the brain generalizes so well.
>
> Coming back to the reviewer’s question, while we may not be able to thoroughly examine this point across many tasks empirically or theoretically given the short rebuttal period (we intend to leave this as future work), our speculation is that bio-plausible rules, when combined with certain architecture, could enable better generalization for specific tasks. This speculation is backed by a recent Neuron paper that investigates how anti-Hebbian rule achieves better OOD generalization than its ML counterpart for familiarity detection task after the network architecture has been meta-learned [1]. The authors nicely explained how the anti-Hebbian learning dynamic is natural for this specific task. However, the study only examined the familiarity detection task. On the other hand, our analysis, by using loss landscape curvature as a tool for studying generalization, is task agnostic. Our analysis, of course, would be limited by to what extent loss landscape curvature can explain generalization, as explained in Discussion.
>
> [1] Danil Tyulmankov, Guangyu Robert Yang, and LF Abbott. Meta-learning synaptic plasticity and memory addressing for continual familiarity detection. Neuron, 110(3):544–557, 2022.

---

> ### Author Response · Authors · 2022-08-02
> **Response to 22dn (1/2)**
>
> We appreciate the reviewer’s supportive comments on the soundness and presentation of the work. As described below, we have built on the reviewer's comments to improve the paper, and we hope we can clarify some of the more contentious points raised. Chiefly, we would like to start by addressing the following:
>
> 1) **“It is well known that bio-plausible methods generally perform worse than SOTA back propagation. This work provides further empirical evidence that this is the case, but I do not find these results very surprising.”**
>
> We agree with the reviewer that it is well known that bio-plausible methods generally perform worse than their ML counterpart. Importantly, we believe the reasons behind this performance gap are not well understood and constitutes an important gap in knowledge. As such, **the main goal of our paper is to make steps toward elucidating some of the reasons behind this phenomenon**. While we contribute a novel systematic evaluation of  generalization gaps for a family of bio-plausible rules, we agree the results of these numerical experiments may not be surprising. This is expected, as they reproduce generally understood cases, and act as the starting point of our main contributions: a mechanistic explanation of the reasons for poorer generalization for existing bio-plausible temporal credit assignment rules. To this end, we leverage theoretical tools from optimization theory and demonstrate, via theorems that match experiments, that bio-plausible rules for temporal credit assignment tend to favor higher curvature minima in the loss landscape. In doing so, we also uncover the importance of distinct sources of variability in bio-plausible rules.
>
> We feel that our contribution sheds light on general mechanisms shared by several proposed bio-plausible learning rules, and that it is a valuable contribution to the existing literature. Importantly, we highlight the novelty in bridging the fields of computational neuroscience, AI, and ML optimization. We take the liberty to quote reviewer eejL who we feel accurately captures the scope of the present study: *“Notably, to my knowledge, this paper is the first to quantitatively explain with high precision the relatively poor generalization performance of biologically-plausible learning rules (see Figure 4).”*
>
> Furthermore our study suggests specific future studies and experiments on learning, and suggests to look beyond learning rules in isolation and consider biological ingredients that can interact with learning rules for better performance (as mentioned in Discussion). We discussed learning rate modulation as one such potential ingredient that may lead to better performance (Appendix Figure 5). In addition, we also believe our paper has additional benefits for the scientific community.
>
> Indeed, our analysis finds that noise from truncation does not affect generalization in the same manner as stochastic gradient noise. This motivates future deep learning (and potentially computational neuroscience) research on how different kinds of noise can impact learning and generalization differently, as mentioned by reviewer dP4V.
>
> More importantly, our work is one of the first that introduces theoretical ML concepts such as loss landscape curvature (and how that affects generalization) to the computational neuroscience community, as mentioned by reviewer dP4V. In doing so, we hope to inspire more future studies that leverage the remarkable progress from the ML community to study the impact of various neural mechanisms on learning, particularly the role of different kinds of noise that occurs in neural systems that may differ from the ones encountered in optimization.
>
> We have now added the above points to the Introduction and Discussion sections to further highlight the contributions of our paper.
>
> **For response part (2/2), please proceed to our comment below.**

---

> ### Author Response · Authors · 2022-08-08
> **Request for discussion**
>
> Given that the author-reviewer discussion period is coming to a close soon, we kindly request the reviewer to let us know if our responses have resolved their concerns, and if there are any other questions that we can address. We are hopeful the reviewer will recognize that our recent work addresses initial concerns. We we are keen to further improve our paper in light of a constructive author-review discussion.

---

### Author Response · Authors · 2022-08-02
**General Response**

We would like to thank all reviewers for their valuable suggestions and constructive feedback. We worked very hard to **address all concerns** raised by the reviewers, and revised the manuscript in light of their suggestions.  As a result, we believe our manuscript is worthy of publication and would be of wide interest to the NeurIPS community. Furthermore, as recognized by some reviewers, this work is one of the first to leverage key theoretical tools from ML optimization theory to better understand biologically plausible learning rules. As NeurIPS is rooted in both theoretical neuroscience and AI, we cannot think of a better venue for this type of cross-disciplinary work, which we hope will seed more follow up.

This review process has been extremely valuable, and led to notable improvements. This is not always the case, and we feel fortunate for attentionate, engaged, and reasonable reviews. In particular, we implemented reviewer q4yu’s comments to significantly improve the clarity of this paper and we would like to extend huge thanks to their valuable comments. We also have added additional simulations in response to reviewer eejL’s specific suggestions on how to sharpen some of our claims even more. We have added explanations to further highlight the key contributions of this work thanks to reviewer 22dn’s comments. Thanks to the great discussion points made by reviewer dP4V’s, we have elaborated our discussions on future work and relationship of work to other important papers in the area. The changes to the manuscript are colored in <span style="color:blue">blue</span> in the present version for easy referencing. Responses to reviewer feedback are provided for each review independently. Overall, we believe that these changes have significantly improved both the content and presentation of our submission.

We also very much appreciate the positive comments regarding the soundness of the paper from reviewers eejL, dP4V and 22dn, which include but are not limited to:

- *“There are no points during the paper which stand out as being unreasonable, and from a scientific method point of view the work appears sound, with each step being justified either from a comp-neuro or ML perspective. Balancing the two topic was done well and I think that is a potentially understated positive for this work"* from reviewer dP4V

- *”The theory is sound and well justified, and the experiments are generally comprehensive”* from reviewer eejL.

Additionally, we would like to thank reviewers dP4V and eejL for recognizing several key contributions of this paper that could motivate exciting future works, such as:

- *“Significant to the comp-neuro community as a step towards introducing some of the more theoretical ML concepts such as loss landscape curvature”* from reviewer dP4V

- *“Generalization in biologically-motivated learning rules has been studied previously. This paper provides surprising new insights into the nature of solutions found by biologically-plausible learning rules. Notably, to my knowledge, this paper is the first to quantitatively explain with high precision the relatively poor generalization performance of biologically-plausible learning rules (see Figure 4)”* from reviewer eejL

---

### Meta-Review · Area_Chair_wsKg · 2022-08-26

**Recommendation:** Accept
**Confidence:** Certain

**Metareview:**

This paper applied ideas about generalization in the ML literature to biologically plausible architectures and learning rules. Especially, it explored links between curvature and generalization in biologically plausible learning.

There was active discussion about this paper, and three reviewers raised their scores during the rebuttal period. All reviewers felt this was a high quality paper, and that the results would be useful for later research. The closest thing to a criticism that came up during discussion was one reviewer describing the paper as a "high quality but incremental addition to the scientific literature."

Based upon the reviews, rebuttal, and reviewer discussion, I recommend paper acceptance. The authors should be sure to update their paper as discussed during the rebuttal period, and based upon the reviewer feedback.

PS -- Links between TBTT and loss surface curvature would also be of interest in learned optimization and meta-learning more broadly, where meta-training is often performed via truncated unrolls of the inner problem.

**Award:**

No

---

### Decision · Program_Chairs · 2022-09-14

Accept